# Synthetic lethality between *VPS4A* and *VPS4B* triggers an inflammatory response in colorectal cancer

Ewelina Szymańska[1],* , Paulina Nowak[1], Krzysztof Kolmus[1], Magdalena Cybulska[2], Krzysztof Goryca[2], Edyta Derezińska-Wołek[3,4], Anna Szumera-Ciećkiewicz[3,4], Marta Brewińska-Olchowik[5], Aleksandra Grochowska[2,6], Katarzyna Piwocka[5], Monika Prochorec-Sobieszek[3,4], Michał Mikula[2] & Marta Miączyńska[1],**

## Abstract

Somatic copy number alterations play a critical role in oncogenesis. Loss of chromosomal regions containing tumor suppressors can lead to collateral deletion of passenger genes. This can be exploited therapeutically if synthetic lethal partners of such passenger genes are known and represent druggable targets. Here, we report that *VPS4B* gene, encoding an ATPase involved in ESCRT-dependent membrane remodeling, is such a passenger gene frequently deleted in many cancer types, notably in colorectal cancer (CRC). We observed downregulation of *VPS4B* mRNA and protein levels from CRC patient samples. We identified *VPS4A* paralog as a synthetic lethal interactor for *VPS4B in vitro* and in mouse xenografts. Depleting both proteins profoundly altered the cellular transcriptome and induced cell death accompanied by the release of immunomodulatory molecules that mediate inflammatory and anti-tumor responses. Our results identify a pair of novel druggable targets for personalized oncology and provide a rationale to develop VPS4 inhibitors for precision therapy of VPS4B-deficient cancers.

**Keywords** CRC; ESCRT; immunogenic cell death; synthetic lethality; VPS4B
**Subject Categories** Autophagy & Cell Death; Cancer

## Introduction

Somatic copy number alterations are a key driving force in malignant transformation (Beroukhim *et al*, 2010). In particular,

chromosomal deletions may affect 35% of the cancer genome (Beroukhim *et al*, 2010). Large deletions may include loss of a tumor suppressor locus, correlating with concomitant loss of hundreds of neighboring protein-coding genes and other regulatory elements. However, it remains elusive whether and how these passenger alterations contribute to cancer development and progression. Nevertheless, passenger alterations may confer a weakness to cancer cells that could be exploited in cancer therapy using synthetic lethality-based approaches.

A synthetic lethal interaction between two genes occurs when perturbation of either gene alone does not alter cell fitness but concomitant perturbation of both genes induces a lethal phenotype. This simple genetic concept underlies well-established studies that characterize functional interactions between two or more genes. Cancer research now exploits this principle to develop new genotype-specific anti-cancer therapeutics (O'Neil *et al*, 2017). Using this approach, PARP inhibitors were introduced as a treatment option for patients with *BRCA1/2*-mutated tumors (Lord & Ashworth, 2017). Continued efforts to develop the synthetic lethality concept for cancer research focus on targeting a non-oncogenic alteration by identifying its second-site synthetically lethal interactor gene. But, developing anti-cancer therapies based on this concept urgently requires identifying and characterizing novel synthetic lethal partners.

Chromosome 18 carries both known and candidate tumor suppressor genes like the well-characterized *DCC* [*Deleted in Colon Cancer Gene*], *SMAD2,* and *SMAD4* (Nguyen & Duong, 2018). Loss of heterozygosity (LOH) at the long arm of this chromosome (18q) can occur in colon (Ogunbiyi *et al*, 1998; Sheffer *et al*, 2009), pancreatic (Sunamura *et al*, 2004), lung (Takei *et al*, 1998), prostate (Kluth *et al*, 2016), and breast cancers (Huiping *et al*, 1998), as well

1  Laboratory of Cell Biology, International Institute of Molecular and Cell Biology, Warsaw, Poland
2  Department of Genetics, Maria Skłodowska-Curie Institute-Oncology Centre, Warsaw, Poland
3  Department of Pathology and Laboratory Medicine, Maria Skłodowska-Curie Institute-Oncology Centre, Warsaw, Poland
4  Department of Diagnostic Hematology, Institute of Hematology and Transfusion Medicine, Warsaw, Poland
5  Laboratory of Cytometry, Nencki Institute of Experimental Biology, Warsaw, Poland
6  Department of Gastroenterology, Hepatology and Clinical Oncology, Medical Center for Postgraduate Education, Warsaw, Poland
   *Corresponding author. Tel: +48 22 597 07 27; E-mail: eszymanska@iimcb.gov.pl
   **Corresponding author. Tel: +48 22 597 07 25; E-mail: miaczynska@iimcb.gov.pl

as head and neck squamous cell carcinoma (Takebayashi et al, 2000). One gene localized on 18q is VPS4B (Vacuolar Protein Sorting 4 Homolog B). Its protein product VPS4B is a ubiquitously expressed type I AAA$^+$ (ATPases Associated with diverse cellular Activities) ATPase (Han & Hill, 2019). It cooperates with its highly homologous paralog VPS4A (encoded by the VPS4A gene on 16q) to disassemble and release the Endosomal Sorting Complex Required for Transport (ESCRT) machinery from intracellular membranes, which enables recycling of ESCRT subunits (Henne et al, 2013; Schoneberg et al, 2017; McCullough et al, 2018).

Endosomal Sorting Complex Required for Transport complexes drive membrane budding and scission at various intracellular compartments (Hurley, 2015) and contribute to a plethora of essential cellular processes, such as endocytic cargo sorting, autophagy, cytokinesis, exovesicle release, and repair of the nuclear envelope, the plasma membrane, and lysosomal membranes (reviewed in refs. Olmos & Carlton, 2016; Christ et al, 2017; Szymanska et al, 2018). In contrast to the well-established roles of ESCRT components in cell biology, their contribution to tumorigenesis, if any, is less documented (Mattissek & Teis, 2014; Alfred & Vaccari, 2016; Gingras et al, 2017) and the underlying molecular mechanisms have been elucidated for only a few cases (Manteghi et al, 2016; Sadler et al, 2018). A large-scale screening for cancer vulnerabilities within the Sanger's Project Score (Behan et al, 2019) and the DRIVE project (McDonald et al, 2017) revealed that some cancer cell lines are very sensitive to perturbed VPS4A expression. The authors of the latter report suggested the existence of a synthetic lethality between VPS4A and VPS4B; however, this hypothesis has not been experimentally verified.

Here, we investigated whether VPS4B expression is perturbed in cancer samples and whether VPS4A is a synthetic lethal partner for VPS4B. Our findings reveal a novel druggable target for further translational research toward personalized therapies for colorectal cancer (CRC) patients.

# Results

## VPS4B expression is deregulated in multiple cancer types, prominently in CRC

To study the extent of genetic changes at the VPS4B locus in different types of cancer, we mined The Cancer Genome Atlas (TCGA). The overview of Pan-Cancer TCGA somatic copy number alteration dataset revealed about a 30% incidence of chromosome 18q LOH at the VPS4B locus (Fig 1A). Further analysis of individual cancer datasets showed frequent VPS4B deletion in several types of cancer with CRC being the most affected (Fig 1B). In line with previous reports on 18q LOH in CRC (Sheffer et al, 2009), an overall incidence of the VPS4B loss in the TCGA CRC dataset was ~70% with bi-allelic deletion estimated at 2% (Fig 1C). In addition, the DNA copy number and mRNA levels of VPS4B were significantly correlated with a Pearson coefficient of 0.75. We next validated changes in VPS4B mRNA abundance using an independent set of CRC samples from our previous studies (Skrzypczak et al, 2010; Mikula et al, 2011). Using qRT–PCR analysis, we confirmed significantly downregulated VPS4B expression, which indicated that its mRNA levels decreased during progression from adenoma to

adenocarcinoma (Fig 1D). In contrast, in the same samples, we detected no change in the level of VPS4A mRNA between normal colon, adenoma, and CRC (Fig EV1A).

To examine whether genetic alterations at the VPS4B locus corresponded to decreased VPS4B protein abundance in CRC, we performed immunohistochemistry (IHC) staining of both paralogs of VPS4 in tissue microarrays covering one hundred pairs of matched human normal colon and treatment-naïve primary CRC samples (Figs 1E and F, and EV1B). We evaluated the microarrays using a semi-quantitative scoring method based on staining intensity (Fig 1E). Antibodies used for IHC staining had been previously tested and approved in The Human Protein Atlas project (https://www.proteinatlas.org/). We confirmed the specificity of the selected antibodies by staining normal human tissues with known high and low protein abundance of both VPS4 paralogs (appendix and muscle, respectively) (Fig EV1C and D). As an additional validation of anti-VPS4B antibody, we confirmed lack of VPS4B staining in mouse xenografts derived from HCT116 human CRC line with both VPS4B alleles inactivated by the CRISPR/Cas9 method (Fig EV1C4 and C5).

When analyzing the tissue microarrays, we found that the staining of VPS4B protein from analyzed pairs of matched cancer patient samples was not as intense as in normal colon (Fig 1F, 3+→3+). In most cases (57%), we observed a slightly decreased VPS4B staining in cancer samples (very intense in normal colon versus medium intense in CRC; 3+→2+). However, we observed a much more prominent decrease of VPS4B staining in CRC (3+→1+) in 39% of tissue pairs and did not detect VPS4B protein in 4% of CRC of tissue samples (3+→0). In contrast, VPS4A staining using the same collection of patient samples demonstrated equally high VPS4A protein abundance in all normal colon and CRC matched pairs (Fig EV1B).

Altogether, our analysis of TCGA databases revealed that the VPS4B allele is frequently lost in many cancer types, prominently in CRC. This genetic alteration correlates with decreased VPS4B mRNA and protein content in cancer tissues that we demonstrated using our collections of CRC patient samples.

## VPS4A is a synthetic lethal partner for VPS4B

VPS4A and VPS4B are the only enzymes among the ESCRT subunits and represent a bottleneck for ESCRT-mediated processes, as no other known ATPases can substitute for their activity. Given our observed aberrations of the VPS4B locus in CRC, we hypothesized that VPS4B and its paralog VPS4A were synthetic lethal interactors. Initially, we chose an in vitro system of a near diploid HCT116 cell line, a human epithelial colorectal carcinoma in which the copy number of VPS4 paralogs is not altered according to the Cancer Cell Line Encyclopedia dataset (CCLE, https://portals.broadinstitute.org/ccle). We assessed its viability upon single RNAi-mediated depletion of VPS4A or VPS4B, as well as simultaneous depletion of VPS4A and VPS4B (referred to as VPS4A+B). We verified high knockdown efficiency and specificity of siRNAs (two or three independent sequences per target) by measuring mRNA and protein abundance of both paralogs (Fig EV2A and B). Despite the sequence similarity between VPS4A and VPS4B, we could efficiently silence the expression of a single VPS4A or VPS4B paralog or both. We revealed that loss of expression of one paralog did not affect the mRNA or protein level of the other (Fig EV2A and B). This suggests that cells with compromised expression of one VPS4 paralog possess no

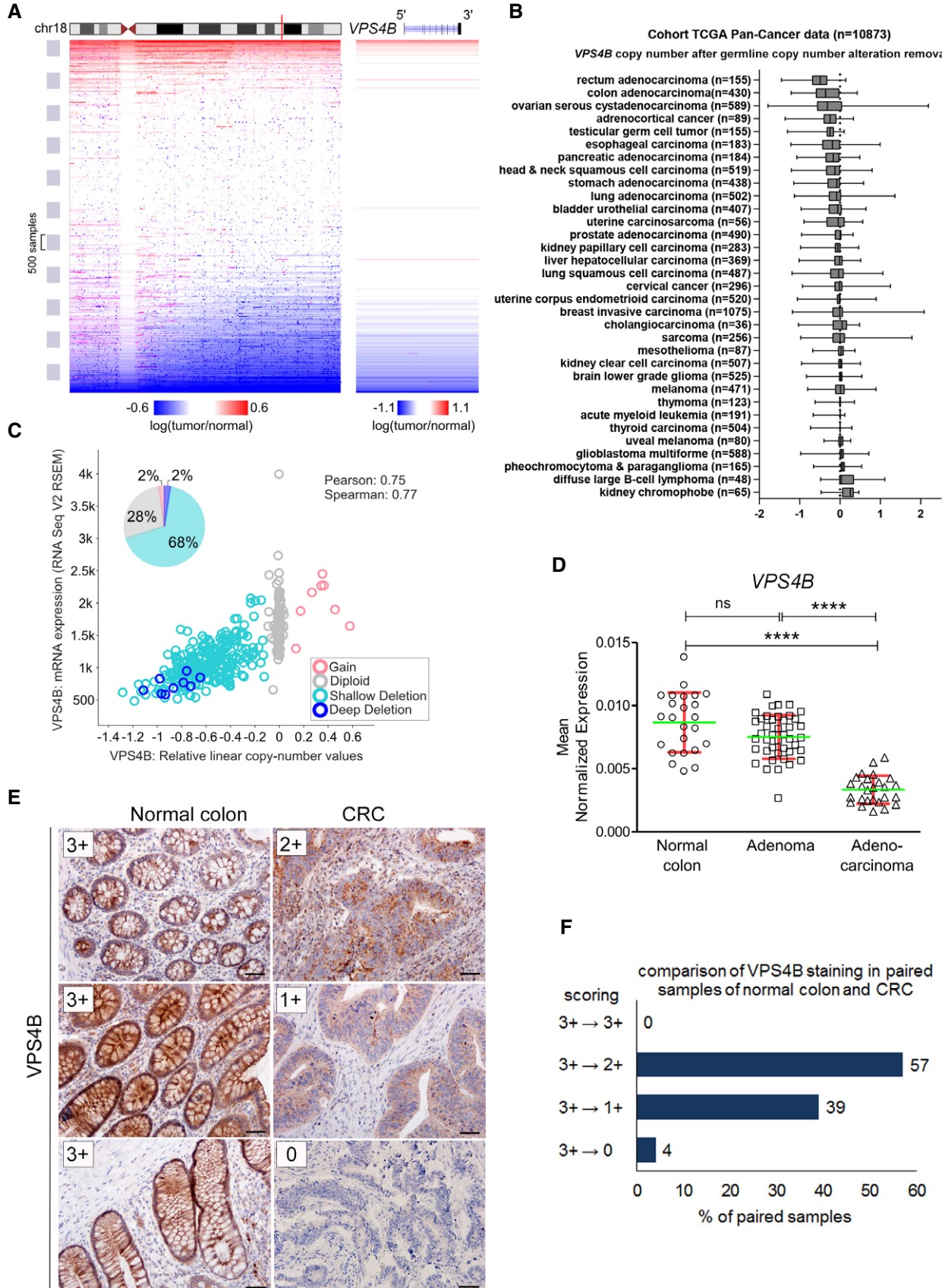

**Figure 1.**

**Figure 1. Expression of *VPS4B* is downregulated in CRC.**

A  Left panel, a scheme of chromosome 18 copy number alterations depicting the distal long arm loss across TCGA Pan-Cancer dataset. Vertical red line indicates the localization of *VPS4B*. Right panel, enlarged fragment of chromosome 18 showing frequent deletions of *VPSB* locus in cancer samples. Deletions are marked in blue, and amplified regions are marked in red. Both panels were generated with UCSC Xena browser (https://www.biorxiv.org/content/10.1101/326470v3).

B  Analysis of *VPS4B* copy number alterations in TCGA Pan-Cancer dataset. Cancer types were sorted according to the mean *VPS4B* copy number after removing germline values. The boxes denote the 25th to 75th percentile range, and the center lines mark the 50th percentile (median). The whiskers reflect the largest and smallest observed values. *VPS4B* copy number alteration data were fetched using UCSC Xena browser.

C  Scatter plot, analysis of *VPS4B* mRNA expression (number of transcripts per million) plotted against *VPS4B* copy number from TCGA CRC patient samples ($n = 376$); plot generated using cBioPortal (Gao *et al*, 2013). Pie chart, summary of all types of *VPS4B* copy number alterations based on the analysis of data from 615 CRC samples deposited in the Colorectal Adenocarcinoma (TCGA, Provisional) dataset on cBioPortal (http://www.cbioportal.org/).

D  qPCR analysis of *VPS4B* mRNA abundance in normal colon, adenoma, and CRC samples. Adenocarcinoma ($n = 26$); adenoma ($n = 42$); normal colon ($n = 24$). Green horizontal bars indicate means, and red whiskers indicate SD. Differences were analyzed using the Kruskal–Wallis test followed by Dunn's multiple comparison *post hoc* test; ns—non-significant ($P \geq 0.05$), ****$P < 0.0001$.

E  Examples of immunohistochemical staining of VPS4B in normal colon and matched CRC samples as an illustration of the scoring system used for the evaluation presented in (F). 3+—very intensive staining, 2+—medium-intensive staining, 1+—weak staining, 0—no staining. Scale bar, 50 μm.

F  Comparative analysis of VPS4B staining performed in 100 pairs of normal colon and matched CRC samples.

Data information: The exact *P*-values can be found in the source data for this figure.
Source data are available online for this figure.

compensatory mechanisms and show no cross-destabilization of the remaining paralog. These results, together with the data showing no correlation between *VPS4A* and *VPS4B* mRNA and protein levels in patient samples (Figs 1D and EV1A; Figs 1F and EV1B, respectively), indicate that the regulation of gene expression and protein stability of both paralogs occurs independently.

Using a short-term viability assay and a long-term colony formation assay, we demonstrated that single depletion of VPS4A or VPS4B did not affect HCT116 growth *in vitro* (Fig 2A and B). However, simultaneous depletion of VPS4A+B significantly reduced HCT116 viability as indicated by decreased proliferation and colony growth. Moreover, we observed similar effects in other CRC cell lines, such as RKO, SW480, and DLD-1, each harboring various cancer driver mutations. We found impaired cell viability of these lines only when expression of both *VPS4* paralogs was silenced simultaneously (Fig 2C).

To confirm the synthetic lethality between *VPS4A* and *VPS4B,* we wished to use a non-engineered cancer cell line with low *VPS4B* expression. To identify such an *in vitro* model, we took advantage of datasets from the Dependency Map (DepMap) portal (https://depmap.org/portal/), updated in the course of revision of this manuscript. This portal systematically catalogs genetic vulnerabilities in human cancer models identified in genome-scale CRISPR/Cas9 and RNAi screens performed within the Broad's Project Achilles (Broad Institute, USA), the Novartis's Project DRIVE (Novartis Institutes for Biomedical Research, Switzerland), and the Sanger's Project Score (Wellcome Sanger Institute, UK). According to DepMap, *VPS4A* and *VPS4B* are "strongly selective genes"; i.e., certain cell lines are vulnerable to the perturbed expression of these genes. Further genetic analyses revealed that cell lines that were the most vulnerable to VPS4A depletion had a decreased copy number of the *VPS4B* gene (Fig 2D). Among them, for our experiments we chose HOP62 and SNU410 cell lines (lung and pancreatic cancer, respectively). These cell lines had been tested in at least two independent screens consistently reaching low *VPS4A* dependency score (CERES or DEMETER2 lower than −0.5 where score of −1 corresponds to the median value of all common essential genes) and relatively high *VPS4B* dependency score (CERES or DEMETER2 higher than −0.5, score of 0 is equivalent to a gene that is not essential; Fig EV2C). We confirmed a decreased *VPS4B* copy number and low VPS4B protein abundance in both selected cell lines (Fig 2E and F). When

expression of *VPS4A* was silenced, these lines exhibited significantly suppressed cell viability and in case of HOP62 also clonal growth (SNU410 cells were not clonogenic) (Fig 2G–I), thus providing an independent validation of the DepMap screening data.

Cumulatively, our data uncovered *VPS4A* and *VPS4B* are novel synthetic lethal interactors, whose mRNA and protein levels undergo independent regulation. The synthetic lethal phenotype demonstrated in various human CRC lines indicates that concomitant loss of *VPS4A* and *VPS4B* expression is deleterious for cells grown *in vitro* independent of their genetic background. Moreover, cancer cells with low expression of *VPS4B* are sensitized to depletion of VPS4A.

**VPS4A depletion inhibits growth of VPS4B-deficient CRC cells *in vivo***

Synthetic lethality observed in *in vitro* settings (such as monolayer growth in cell culture) may occasionally be modified or even rescued in a more complex *in vivo* environment where tumor grows in a three-dimensional space and interact with stroma (Ryan *et al*, 2018). So, we used a mouse xenograft model to reproduce the synthetic lethality between *VPS4A* and *VPS4B*. For this long-term analysis, we generated a CRISPR/Cas9 engineered HCT116 line with knockout of *VPS4B* (HCT116 *VPS4B$^{-/-}$*). Sequence analysis of PCR products covering *VPS4B* revealed a frameshift mutation in exon 3 of two clones: 1C5 and 2B3 (Fig EV3A). Further IHC and immunoblotting analysis confirmed lack of VPS4B protein in both clones (Figs EV1C4, C5, and EV3A). In agreement with the data on RNAi-mediated silencing of *VPS4B* (Fig 2A and B), neither growth rate *in vitro* nor *in vivo* (as xenografts in mice) was affected in HCT116 VPS4B$^{-/-}$ clones in comparison with the parental HCT116 VPS4B$^{+/+}$ line (Fig EV3B and C). Importantly, we could recapitulate the synthetic lethality between *VPS4A* and *VPS4B* in an *in vitro* assay upon transfection of HCT116 *VPS4B$^{-/-}$* cells by siRNA targeting *VPS4A* (Fig EV3D).

Next, we transduced HCT116 *VPS4B$^{-/-}$* clone 2B3 by lentiviral constructs bearing doxycycline-inducible shRNAs targeting *VPS4A* mRNA (HCT116 *VPS4B$^{-/-}$* shVPS4A#1, #2, #3) or generated control cell lines using constructs bearing non-targeting shRNA sequences (HCT116 *VPS4B$^{-/-}$* shCTRL#1, #2). Of the three shVPS4A sequences tested, sequence #1 was the most efficient in decreasing VPS4A

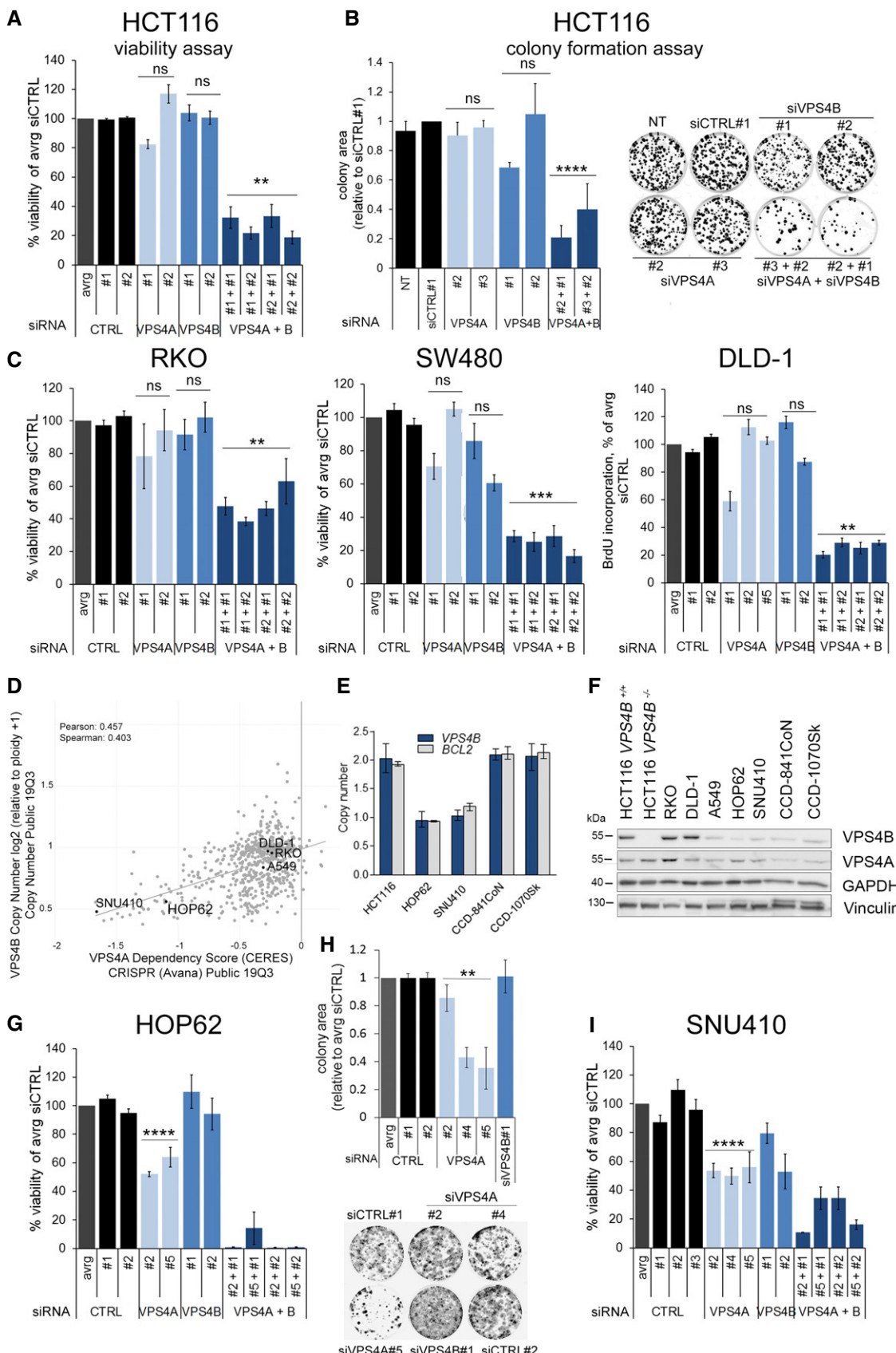

**Figure 2.**

**Figure 2. Synthetic lethality between *VPS4A* and *VPS4B* inhibits growth of CRC lines *in vitro*.**

A   Analysis of viability of HCT116 cells assessed 96 h after transfection with independent non-targeting siRNA (two different duplexes used, siCTRL#1 or #2) or targeting *VPS4A* (duplexes #1 or #2), *VPS4B* (duplexes #1 or #2), or both *VPS4* (various combinations of siVPS4A+siVPS4B duplexes). Data are means of three independent experiments ± SEM. All values were normalized, averaged (avrg) viability of siCTRL#1- and #2-transfected cells was set as 100%, the Kruskal–Wallis test followed by Dunn's multiple comparison *post hoc* test; ns—non-significant ($P \geq 0.05$), **$P < 0.01$.

B   Left panel, analysis of clonogenic growth of HCT116 cells assessed 15 days after transfection with non-targeting (siCTRL#1), *VPS4A*- or/and *VPS4B*-targeting siRNAs used in various duplex combinations as indicated. Right panel, images of HCT116 clones taken at the day of clonogenic growth assessment. Data are means of three independent experiments ± SEM. NT—non-transfected. All values were normalized, and clonogenic growth of siCTRL#1-transfected cells was set as 1. One-sample *t*-test; ns—non-significant ($P \geq 0.05$), ****$P < 0.0001$.

C   Viability of RKO and SW480 cells assessed as in (A), growth of DLD-1 cells assessed in BrdU incorporation assay 96 h after siRNA transfection. Various independent non-targeting (siCTRL#1 or #2) and *VPS4A*- or *B*-targeting siRNA duplexes were used (#1, #2, #5 or #1, #2, respectively). All values were normalized, and averaged (avrg) viability of siCTRL#1- and #2-transfected cells was set as 100%. Data are means of three independent experiments ± SEM. The Kruskal–Wallis test followed by Dunn's multiple comparison *post hoc*; ns—non-significant ($P \geq 0.05$), **$P < 0.01$, ***$P < 0.001$.

D   *VPS4B* copy number and dependency scores of selected cancer cell lines obtained in CRISPR/Cas9 and RNAi screens deposited in the DepMap portal (https://depmap.org/portal/).

E   *VPS4B* copy number status estimated across different cell lines using TaqMan assay. The error bars represent the minimal and maximal copy number in a given triplicate readout. The *BCL2* gene localized on 18q in the close vicinity to *VPS4B* was analyzed as a control.

F   VPS4B and VPS4A protein abundance in selected normal and cancer cell lines analyzed by immunoblotting. Vinculin and GAPDH were used as loading controls.

G   Analysis of viability of HOP62 lung cancer cells assessed 144 h after transfection with independent non-targeting siRNA (two different duplexes used, siCTRL#1 or #2) or targeting *VPS4A* (duplexes #2 or #5), *VPS4B* (duplexes #1 or #2), or both *VPS4* (various combinations of siVPS4A+siVPS4B duplexes). Data are means of four independent experiments ± SEM. All values were normalized, and cell viability of averaged (avrg) siCTRL#1- and #2-transfected cells was set as 100%. Two-tailed unpaired *t*-test; ****$P < 0.0001$.

H   Top panel, analysis of clonogenic growth of HOP62 cells assessed 14 days after transfection with various siRNA duplexes as indicated. Bottom panel, images of HOP62 clones taken at the day of clonogenic growth assessment. Data are means of four independent experiments ± SEM. All values were normalized, and colony area of averaged (avrg) siCTRL#1- and #2-transfected cells was set as 1. The Mann–Whitney *U*-test; **$P < 0.01$.

I   Analysis of viability of SNU410 pancreatic cancer cells assessed 168 h after transfection with independent non-targeting siRNA (three different duplexes used, siCTRL#1, #2, or #3) or targeting *VPS4A* (duplexes #2, #4, or #5), *VPS4B* (duplexes #1 or #2), or both *VPS4* (various combinations of siVPS4A+siVPS4B duplexes). Data are means of three independent experiments ± SEM. All values were normalized, cell viability of averaged (avrg) siCTRL#1-, #2-, and #3-transfected cells was set as 100%. Two-tailed unpaired *t*-test; ****$P < 0.0001$.

Data information: The exact *P*-values can be found in the source data for this figure.
Source data are available online for this figure.

protein level upon doxycycline induction (Fig EV3E), thus leading to the synthetic lethality that limited viability of HCT116 *VPS4B*$^{-/-}$ cells *in vitro* (Fig EV3F). So, we selected HCT116 *VPS4B*$^{-/-}$ shVPS4A#1 cells (hereafter referred to as HCT116 *VPS4B*$^{-/-}$ shVPS4A) for further *in vivo* studies.

We decided to induce synthetic lethality in mice after tumors reached a certain size rather than at the time of xenotransplantation, since this approach better mimics a potential therapeutic intervention in patients with VPS4B-deficient CRC tumors. So, we injected subcutaneously $5 \times 10^6$ of HCT116 *VPS4B*$^{-/-}$ shVPS4A cells to immunocompromised NU/J mice. When tumors reached at least 150 mm$^3$, we divided the animals into two groups in which one received doxycycline in drinking water to induce shRNA expression in the tumor cells (Fig 3A). Using this approach, we demonstrated that the induction of shVPS4A expression in xenografted HCT116 *VPS4B*$^{-/-}$ cells caused a significant retardation in tumor growth (Fig 3B). We also confirmed decreased VPS4A protein abundance in all but one xenograft samples derived from doxycycline-fed mice (Fig 3C). We suspect that unchanged VPS4A protein level in this sample (#6) might result from a limited doxycycline availability for the growing xenograft (e.g., the mouse had drunk less water or the tumor was poorly vascularized). We further excluded a toxic effect of doxycycline alone on tumor growth or mouse metabolism because the body weight of all mice and the growth of xenografted HCT116 *VPS4B*$^{-/-}$ shCTRL#1 cells were similar in doxycycline-treated and untreated animals (Figs 3D and EV3G).

Thus, we confirmed that depleting VPS4A protein in CRC cells with compromised *VPS4B* expression inhibits tumor growth in mouse xenografts.

**Transcriptome of VPS4A+B-depleted cells exhibits upregulated gene expression in inflammatory responses and programmed cell death pathways**

We predicted that the lethal phenotype of VPS4A+B-depleted cells might result from simultaneous perturbation of several ESCRT-dependent processes. Among them, we analyzed endocytosis and cell cycle progression in HCT116 cells. In line with previous reports (Bishop & Woodman, 2000; Vietri *et al*, 2015; Mierzwa *et al*, 2017), we demonstrated that concomitant depletion of VPS4A+B perturbed the morphology (Fig EV4A) and function of the endocytic system (assayed by transferrin uptake, Fig EV4B) and caused G2/M cell cycle arrest due to impaired cytokinesis (Fig EV4C).

To investigate any further mechanistic consequences of VPS4 depletion on cellular homeostasis, we performed RNA sequencing (RNA-Seq) in HCT116 cells. We used two independent siRNA sequences per target to silence *VPS4* paralogs individually or in combination, compared to non-transfected cells and cells transfected with non-targeting siRNA (siCTRL#1). In our data analysis, we focused on genes whose expression was above 1.5-fold and adjusted *P*-value < 0.05 in all comparisons of two on-target siRNAs with two control conditions (Appendix Fig S1A). We observed the strongest changes in gene expression exerted by combined silencing of *VPS4A* + *B* that induced transcription of 587 genes (Appendix Fig S1A, Dataset EV1). Knockdown of VPS4B alone upregulated expression of 58 genes (Appendix Fig S1A, Dataset EV2), while silencing of *VPS4A* did not induce any gene transcription (Appendix Fig S1A). These data correlate well with unchanged growth properties of cells with single depletion of one paralog (Fig 2A and B).

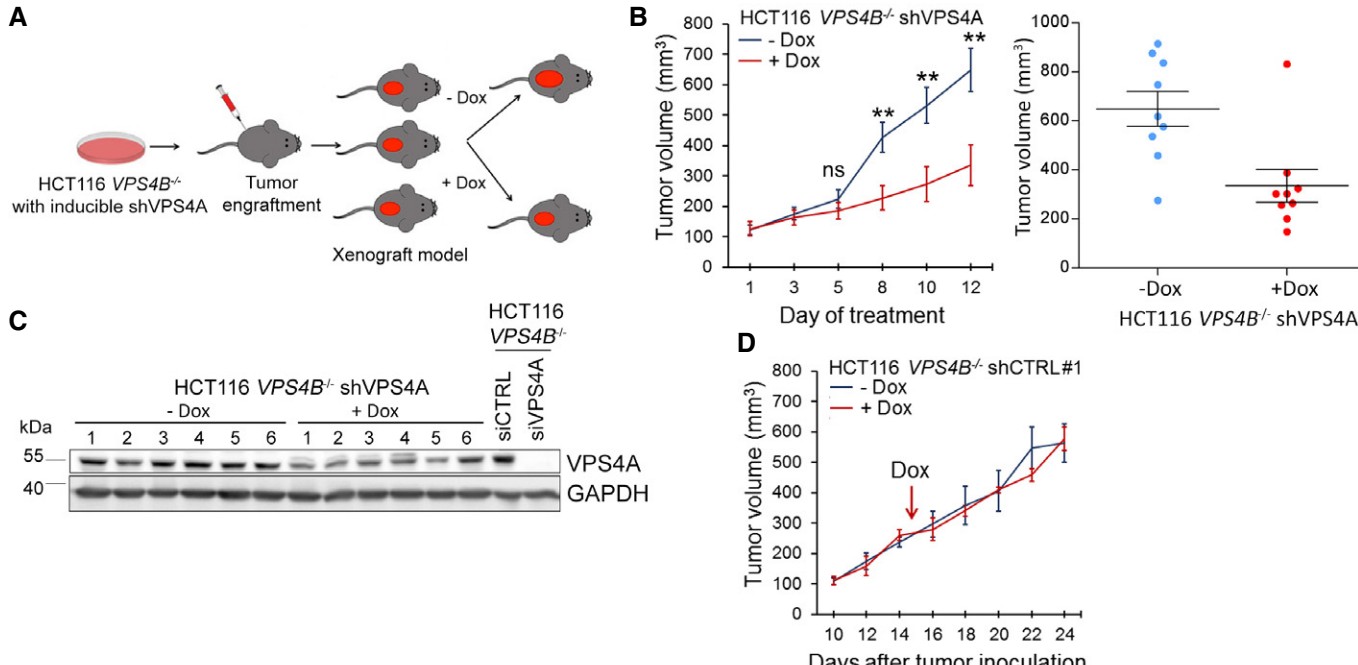

**Figure 3. Synthetic lethality between *VPS4A* and *VPS4B* inhibits growth of CRC cells in a mouse xenograft model.**

A  Schematic illustration of xenograft experiments with HCT116 *VPS4B*$^{-/-}$ cells having doxycycline (Dox)-inducible expression of shRNA targeting *VPS4A*.

B  Left panel, growth of HCT116 *VPS4B*$^{-/-}$ shVPS4A cells as xenografts in mice in the presence or absence of doxycycline. Day 1 indicates the first day of doxycycline administration. $n = 9$ for each group, each mouse bearing one tumor, $\pm$ SEM. Two-tailed unpaired $t$-test; ns—non-significant ($P \geq 0.05$), **$P < 0.01$. Right panel, scatter plot representing end-point volumes of single xenografts. Bars represent means $\pm$ SEM.

C  Immunoblotting analysis of VPS4A abundance in xenograft samples from untreated and doxycycline-treated mice (6 separate xenograft samples for each group). Lysates of HCT116 *VPS4B*$^{-/-}$ cells transfected with control or *VPS4A*-targeting siRNA marked VPS4A protein detection. GAPDH served as a loading control.

D  Growth of HCT116 *VPS4B*$^{-/-}$ shCTRL#1 cells as xenografts in mice in the presence or absence of doxycycline. The arrow indicates the first day of doxycycline (Dox) administration. $n = 2$ mice for each group, each mouse bearing two tumors, $\pm$ SEM.

The exact *P*-values can be found in the source data for this figure.

Source data are available online for this figure.

To analyze the transcriptomic effects of simultaneous depletion of VPS4A+B, we first performed gene ontology (GO) analysis of biological processes taking into account the 587 upregulated genes after combined silencing of *VPS4A + B*. Our data revealed that among the top 15 gene signatures were inflammatory response (GO:0006954) ($P < 0.0001$) and positive regulation of programmed cell death (GO:0043068) ($P = 0.0029$) (Fig 4A). In parallel, we performed gene set enrichment analysis (GSEA) for genes expressed in individual comparisons of combined VPS4A+B depletion versus control conditions. Consistently with the annotations found in GO analysis of biological process, GSEA revealed the presence of inflammatory responses and positive regulation of programmed cell death signatures in each comparison of on-target siRNAs versus control condition (Appendix Fig S1B, for the Normalized Enrichment Score (NES) and False Discovery Rate (FDR) values refer to Appendix Table S1). Hierarchical clustering of all samples on a set of genes linked to the inflammatory response and the cell death signature demonstrated that the branch of VPS4A+B-depleted samples was clearly distinct from the remaining samples (Fig 4B, Dataset EV3). The inflammatory response heatmap contained a distinct cluster of genes encoding several cytokines, such as *CXCL2, CXCL8, IL18,* and NF-κB signaling components, like *NFKBIA, TNFAIP3,* and *BIRC3* whose expression was strongly upregulated

after combined VPS4A+B depletion (Fig 4B). Consistent with the cell death signature, we found increased transcription of *BAK1, BIK,* and *BCL2L11* after combined VPS4A+B knockdown (Fig 4B).

In order to determine the signaling pathways associated with inflammatory and cell death signatures after combined *VPS4A + B* silencing, we conducted a pathway network analysis using the Reactome Database. As shown in Fig 4C, our analysis of differentially expressed genes indicated an enrichment of annotations related to signaling of interleukins and death receptors, as well as regulation of necroptosis.

Collectively, VPS4A+B knockdown profoundly affects gene expression patterns, predominantly linked to the induction of inflammatory response and programmed cell death, in contrast to single depletion of VPS4 proteins that had little or no effects on transcription. In-depth data interrogation suggests apoptosis initiated via death domain-containing receptors and necroptosis as putative mechanisms responsible for cell death.

## VPS4A+B depletion induces diverse cell death execution programs

To further dissect which cell death pathway(s) contributed to the *VPS4A + B* synthetic lethal phenotype, we performed

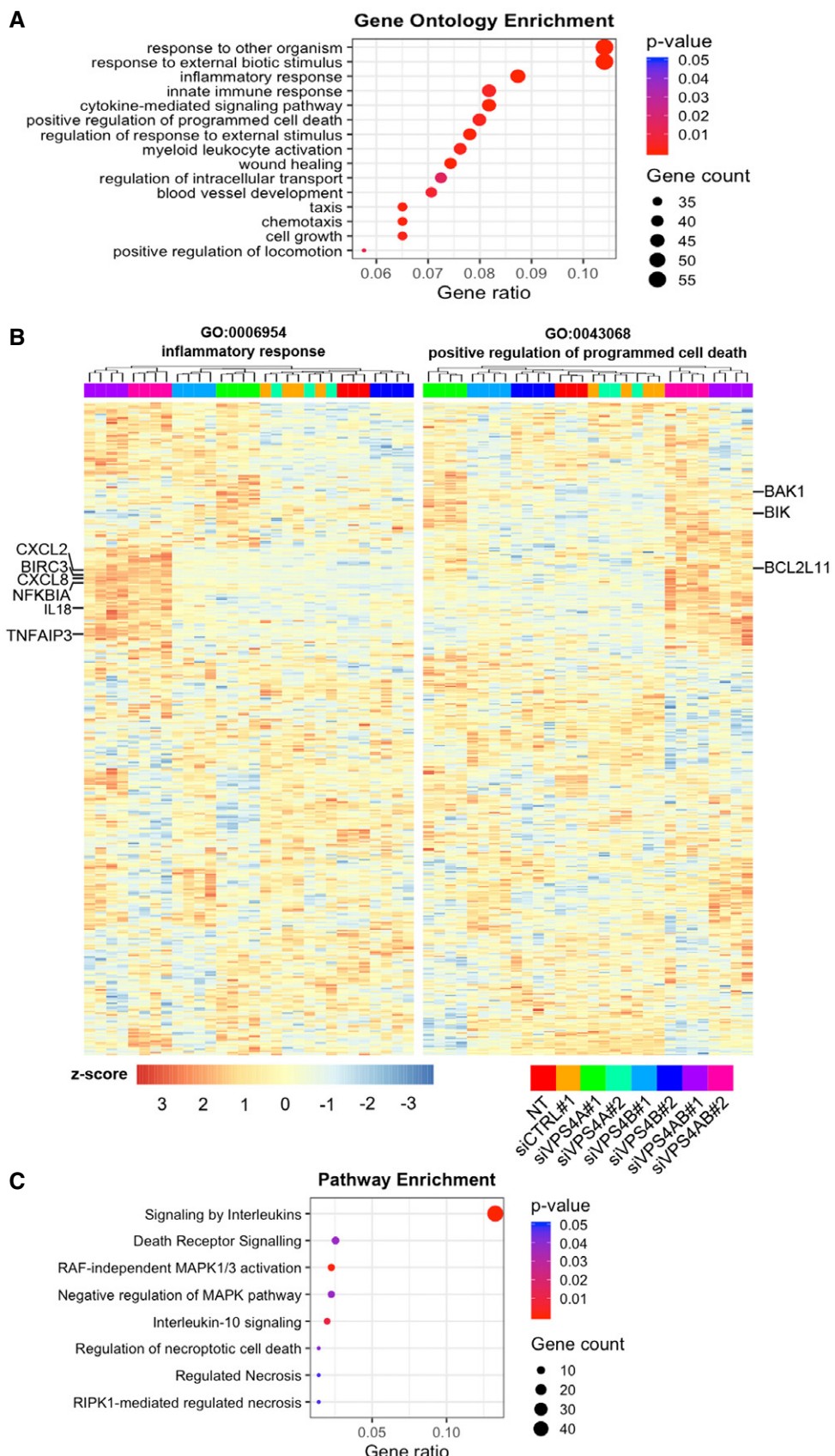

Figure 4.

**Figure 4. Combined knockdown of VPS4 proteins in HCT116 cells induces alterations in the transcriptome.**

A Gene ontology (GO) analysis of biological processes for transcriptionally upregulated genes (≥ 1.5-fold; adjusted *P*-value < 0.05) after combined *VPS4A+B* silencing using the enrichGO function from clusterProfiler.

B Heatmap visualizing expression of genes related to inflammatory response (left panel) and positive regulation of cell death (right panel) generated from the GO analysis of biological processes across different transfection conditions with at least three biological replicates.

C Selected pathways related to inflammatory response and programmed cell death among transcriptionally upregulated genes after combined *VPS4A+B* silencing were identified using the enrichPathway function from ReactomePA.

immunoblotting analysis of unprocessed and cleaved forms of caspases, as hallmarks of apoptotic cell death. We observed that silencing of *VPS4A* expression in HCT116 *VPS4B$^{-/-}$* cells caused cleavage of caspase 8 and caspase 9, two initiator enzymes of extrinsic and intrinsic branches of apoptosis, respectively. Activation of these caspases led to the cleavage of caspases 3 and 7 (executioner enzymes common for both branches) and their downstream substrate PARP-1 (Fig 5A and B). We also confirmed apoptotic cell death in mouse xenografts having both VPS4 proteins simultaneously depleted (Fig 5C). However, inhibition of caspase activity by 20 μM Q-VD-Oph (a pan-caspase inhibitor) only partially improved the viability of VPS4A+B-depleted cells *in vitro* (Fig 5D). We verified that this inhibitor concentration efficiently blocked activation of caspases 3 and 7 in HCT116 cells (Appendix Fig S2A). This suggested that a caspase-independent cell death program could operate in parallel to caspase-mediated apoptosis.

In line with these observations, the RIPK1-regulated cell death pathway showed transcriptional upregulation in VPS4A+B-depleted cells (Fig 4C). However, classical RIPK1- and RIPK3-dependent necroptosis (the second, after apoptosis, most common type of programmed cell death) appears unlikely to operate in HCT116 cells as they do not express RIPK3 (Koo *et al*, 2015; Moriwaki *et al*, 2015), a key enzyme in the necroptosis pathway. However, HCT116 cells possess RIPK1 (Moriwaki *et al*, 2015). A recent report indicates RIPK1 can mediate noncanonical caspase- and RIPK3-independent cell death (Mihaly *et al*, 2017). Indeed, inhibition of RIPK1 activity by necrostatin-1 improved the viability of VPS4A+B-depleted cells, although only to a limited degree similar to the effect of caspase inhibition by Q-VD-Oph (Fig 5D). However, combined treatment with Q-VD-Oph and necrostatin-1 further rescued cell viability (Fig 5D). This finding suggests that both cell death pathways, caspase-dependent apoptosis and the caspase-independent RIPK1-mediated pathway, may operate in parallel upon synthetic lethal perturbation of VPS4A and VPS4B.

**Loss of VPS4A and VPS4B proteins activates NF-κB signaling**

Transcriptome analysis of VPS4A+B-depleted HCT116 cells revealed that cell death induced by synthetic lethality is accompanied by upregulated expression of inflammatory response genes with many linked to NF-κB signaling (Fig 4A and B). Immunoblotting analysis of lysates of VPS4A-depleted HCT116 *VPS4B$^{-/-}$* cells grown *in vitro* and *in vivo* confirmed activation of canonical and noncanonical branches of the NF-κB pathway, marked by IκBα and RelA phosphorylation, and cleavage of p100 to p52, respectively (Fig 5A–C). Transcription factors from the NF-κB family are generally regarded as anti-apoptotic; however, their pro-apoptotic activities can occasionally occur in cells stressed by specific stimuli (Perkins & Gilmore, 2006; Radhakrishnan & Kamalakaran, 2006; Strozyk *et al*, 2006). So, we

investigated the role of the NF-κB pathway in executing cell death in VPS4A+B-depleted cells. We demonstrated that knockdown of RelA in VPS4A+B-depleted cells reduced caspase activation (Appendix Fig S2B) and prevented cell death and detachment, as observed in phase contrast microscopy (Appendix Fig S2C). These results suggest that intracellular stress conditions generated by *VPS4A + B* synthetic lethality activate pro-apoptotic functions of RelA.

Overall, we conclude that the induction of cell death following VPS4A+B loss is accompanied by activation of NF-κB signaling and its canonical branch to promote caspase activation.

**Cell death evoked by *VPS4A+B* synthetic lethality triggers release of immunomodulatory molecules and damage-associated molecular patterns**

Recent studies indicate that some detrimental stimuli (mostly chemotherapeutics) can induce a particular form of programmed cell death called immunogenic cell death (ICD), which involves exposure or secretion by dying cells of a specific set of damage-associated molecular patterns (DAMPs), among them ATP, HMGB1, or calreticulin (Kepp *et al*, 2014; Wang *et al*, 2018). They are recognized by cells mediating innate and adaptive immunity that leads not only to the elimination of cell remnants but, more importantly, establishing anti-tumor immunological memory. To examine whether simultaneous silencing of both *VPS4* paralogs stimulates release of DAMPs typical for ICD, we analyzed secretion of ATP and HMGB1. Indeed, VPS4A depletion in HCT116 *VPS4B$^{-/-}$* cells increased secretion of ATP (Fig 6A) and HMGB1 (Fig 6B) to the medium. Moreover, many HCT116 *VPS4B$^{-/-}$* cells exposed calreticulin on their surface upon VPS4A depletion, as we demonstrated by fluorescence microscopy (Fig 6C) and confirmed by flow cytometry (Fig 6D, Appendix Fig S3). These data suggest that perturbation of VPS4A activity in VPS4B-deficient cancer cells can evoke release of immunomodulatory DAMPs.

Finally, we examined whether DAMPs released by VPS4A+B-deficient cancer cells could elicit paracrine effects on primary immune cells. In these experiments, we used CT-26 cells, a mouse colon carcinoma line that was transiently transfected by siRNA targeting mouse *Vps4a* and *Vps4b* (Fig EV5A and B). First, we confirmed that silencing of *Vps4a* and *Vps4b* in mouse colon cancer cells recapitulated the synthetic lethality observed in human CRC cells, such that it was accompanied by activation of caspase 7 and canonical NF-κB signaling leading to cell death (Fig EV5B and C). We then tested the effects of DAMPs on immune cells by incubating primary bone marrow-derived macrophages in conditioned medium collected from Vps4a+b-depleted CT-26 cells. Macrophage activation toward an anti-tumor and pro-inflammatory M1 state, or a pro-tumor and anti-inflammatory M2 state was analyzed by measuring expression of relevant marker genes by qRT–PCR. Depletion of any

single paralog did not activate macrophages toward M1 or M2. However, DAMPs released from cells with Vps4a+b knockdown prompted macrophage activation toward the M1 phenotype (Fig 6E), as the expression of all four M1 markers analyzed was significantly increased.

In summary, we revealed that *VPS4A + B* synthetic lethal phenotype of dying cells is accompanied by exposure and secretion of molecules governing inflammatory and anti-tumor response.

## Discussion

Developing personalized oncology requires identifying novel targets for selective killing of genetically and phenotypically diverse tumor cells. Moreover, we must understand the cellular functions of these targets and the biological consequences of their perturbation to evaluate the effectiveness and side effects of tailored therapies. Here, by demonstrating the synthetic lethal interaction between two

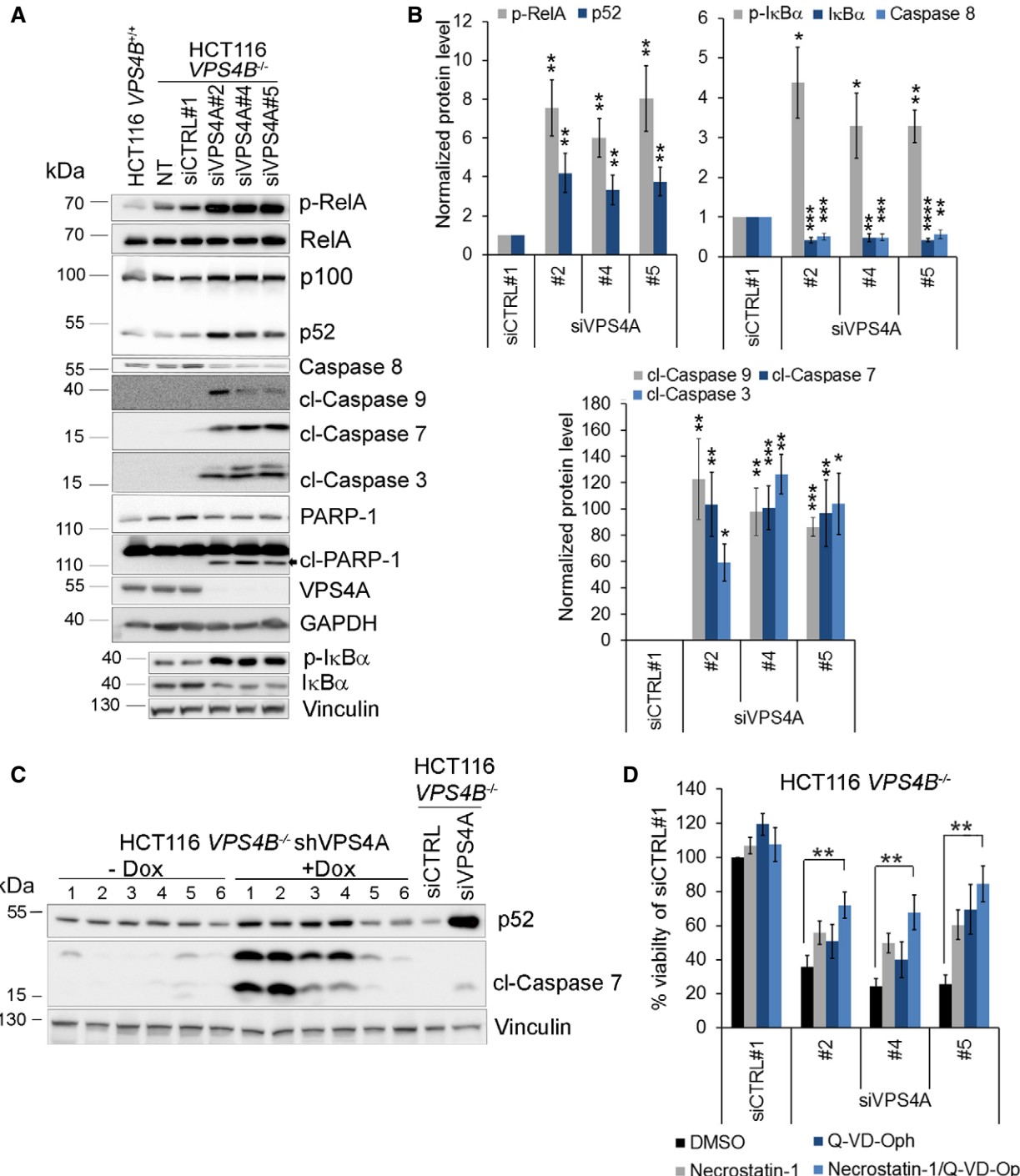

**Figure 5.**

**Figure 5. Combined VPS4A+B depletion induces NF-κB signaling and caspase-dependent and caspase-independent cell death pathways.**

A  Immunoblotting analysis of the canonical and noncanonical branches of the NF-κB pathway and mediators of caspase-dependent cell death. Lysates of HCT116 *VPS4B*$^{-/-}$ cells were collected 66 h after transfection with siRNA (siCTRL#1 or different siVPS4A duplexes: #2, #4, or #5). Lysates of HCT116 *VPS4B*$^{+/+}$ and non-transfected HCT116 *VPS4B*$^{-/-}$ cells were used to monitor the basal pathway activity. Representative blot from 10 experiments is shown. NT—non-transfected; p-RelA —phospho-RelA; p-IκBα—phospho-IκBα; cl—cleaved caspases or PARP-1. GAPDH or vinculin served as loading controls.

B  Densitometry analysis of the abundance of the indicated proteins based on immunoblot images as shown in (A). Data are means of 10 (phospho-Rel and cleaved caspase 7), nine (p52, caspase 8 and cleaved caspase 9), seven (phosphorylated and total IκB), or five (cleaved caspase 3) independent experiments. Error bars are SEM. Statistical significance was assessed using the following tests: one-sample *t*-test (caspase 8, cleaved caspases 9, 7, and 3, total and phospho-IκBα) or Wilcoxon signed rank test (phospho-RelA and p52); *$P < 0.05$, **$P < 0.01$, ***$P < 0.001$, ****$P < 0.0001$.

C  Immunoblot showing induction of the NF-κB pathway (p52) and apoptosis (cleaved caspase 7) in xenograft samples described in Fig 3C. cl—cleaved caspase. Vinculin was used as a loading control.

D  Analysis of the impact of RIPK1 inhibitor (necrostatin-1) or pan-caspase inhibitor (Q-VD-Oph) on cell viability of HCT116 *VPS4B*$^{-/-}$ cells transfected with siRNA (non-targeting siCTRL#1 or different siVPS4A duplexes: #2, #4, or #5). Cell viability was assessed 72 h after siRNA transfection. Necrostatin-1 (50 μM), Q-VP-Oph (20 μM), or vehicle were added to the medium 48 h before viability assessment. Data are means of five independent experiments ± SEM. All values were normalized, and viability of siCTRL-transfected and vehicle-treated cells was set as 100%. Two-tailed unpaired *t*-test; **$P < 0.01$.

Data information: The exact p-values can be found in the source data for this figure.
Source data are available online for this figure.

ubiquitously expressed human paralogs *VPS4A* and *VPS4B*, we uncovered a novel therapeutic target to treat patients bearing VPS4B-deficient cancers, for example, CRC used as a model in our study. We also identify the molecular consequences of perturbing VPS4 in cancer cells and propose that inflammatory cell death triggered by VPS4A+B depletion can evoke an anti-tumor response (Fig 7). Thus, our study contributes to both cell biology and cancer therapy research.

In humans, *VPS4A* and *B* paralogs are located on separate chromosomes (16q and 18q, respectively). Their protein products with 81% sequence identity likely co-operate by forming hetero-oligomers involved in ATP-dependent disassembly of ESCRT-III complexes, a final step in membrane severing in key processes like endocytic sorting, cytokinesis, and membrane repair (Scheuring *et al*, 2001; Henne *et al*, 2013; Schoneberg *et al*, 2017; McCullough *et al*, 2018). Still, it remained unknown to what extent both VPS4 proteins simply duplicate each other's functions or whether they also possess paralog-specific roles. Our data argue that VPS4A and VPS4B are functionally mostly redundant, but the loss of both has deleterious consequences for the cell. However, the different chromosomal localization of *VPS4* genes could favor independent regulation of their expression in normal or pathological tissues. Changes in mRNA or protein abundance of *VPS4A* or *VPS4B* were reported in hepatoma, breast, and non-small-cell lung cancer (Lin *et al*, 2012; Liu *et al*, 2013; Jiang *et al*, 2015; Wei *et al*, 2015). These studies suggested VPS4A acted as a tumor suppressor (Wei *et al*, 2015), while VPS4B exhibited pro- or anti-oncogenic activities depending on the tumor type, which could reflect the multiple cellular functions of VPS4 proteins (Lin *et al*, 2012; Liu *et al*, 2013; Jiang *et al*, 2015). However, a systematic analysis of genomic rearrangements of *VPS4* loci and abundance of *VPS4* mRNAs and proteins across tumor types has not been reported.

By exploring TCGA datasets, we found here that the *VPS4B* gene undergoes deletion as a part of 18q in many cancer types (Fig 1). We observed the highest frequency of shallow (possibly heterozygous) and deep (possibly homozygous) deletions of *VPS4B* in CRC. Accordingly, we discovered that *VPS4B* expression is progressively downregulated from adenoma to adenocarcinoma and confirmed a decreased abundance of VPS4B protein in treatment-naïve primary CRC samples. Our data go in line with the very recently published proteogenomic analysis of colon tumors (Vasaikar *et al*, 2019).

Additionally, our findings suggest that *VPS4B* downregulation occurs early in tumorigenesis and undergoes positive selection during development of primary CRC. However, silencing of *VPS4B* expression or its bi-allelic knockout did not improve growth of CRC cell lines *in vitro* or *in vivo*. In addition, analysis of the transcriptome of CRC cells upon VPS4B depletion showed weakly upregulated expression of < 60 genes, among which we did not find known tumorigenesis drivers. Based on these data, we conclude that CRC-associated downregulation of *VPS4B* does not reflect its own tumor suppressor function but rather represents a passenger alteration co-existing with concomitant loss of the neighboring 18q-located tumor suppressors, such as *DCC*, *SMAD2*, and *SMAD4* (Nguyen & Duong, 2018). 16q-located *VPS4A* was consistently not affected in CRC. We confirmed this by showing its unaltered mRNA and protein abundance both in cancer and healthy tissue. Nevertheless, we cannot exclude that *VPS4B* downregulation may be beneficial for cells of advanced-stage CRC, as RNAi-mediated knockdown of VPS4B may promote resistance of multiple melanoma and breast cancer cells to chemotherapeutics (Lin *et al*, 2012; Tang *et al*, 2015). Further experiments will need to clarify this interesting issue.

Importantly, our demonstration of synthetic lethality between druggable *VPS4* paralogs provides a rationale to develop novel therapies targeting VPS4A activity in cancers with 18q deletion, such as CRC. Although synthetic lethality has been long proposed as a promising strategy to target genetic defects in tumors, its clinical utility is unfortunately still limited. Since many identified synthetic lethal interactions demonstrate incomplete penetrance, they are valid only for a fraction of tumor cells with a specific genetic background and are abrogated by other mutations that may appear in a heterologous population of tumor cells (Shen *et al*, 2017; Ryan *et al*, 2018). Together with other reports, our findings reaffirm prioritizing synthetic lethal partners with highly penetrant effects for further clinical assessment. We believe that *VPS4A* and *VPS4B* are highly penetrant interactors. First, *VPS4A* is the only paralog that can overtake the function of *VPS4B* and vice versa. Second, proteins encoded by these paralogs are likely direct physical interactors (Scheuring *et al*, 2001; Huttlin *et al*, 2015, 2017) and participate in multiple pathways essential for growth across many cell types (Christ *et al*, 2017). Third, we demonstrated that various genetic backgrounds of cancer cell lines did not reverse the synthetic lethality between *VPS4* paralogs (Fig 2). Finally, we showed that the synthetic

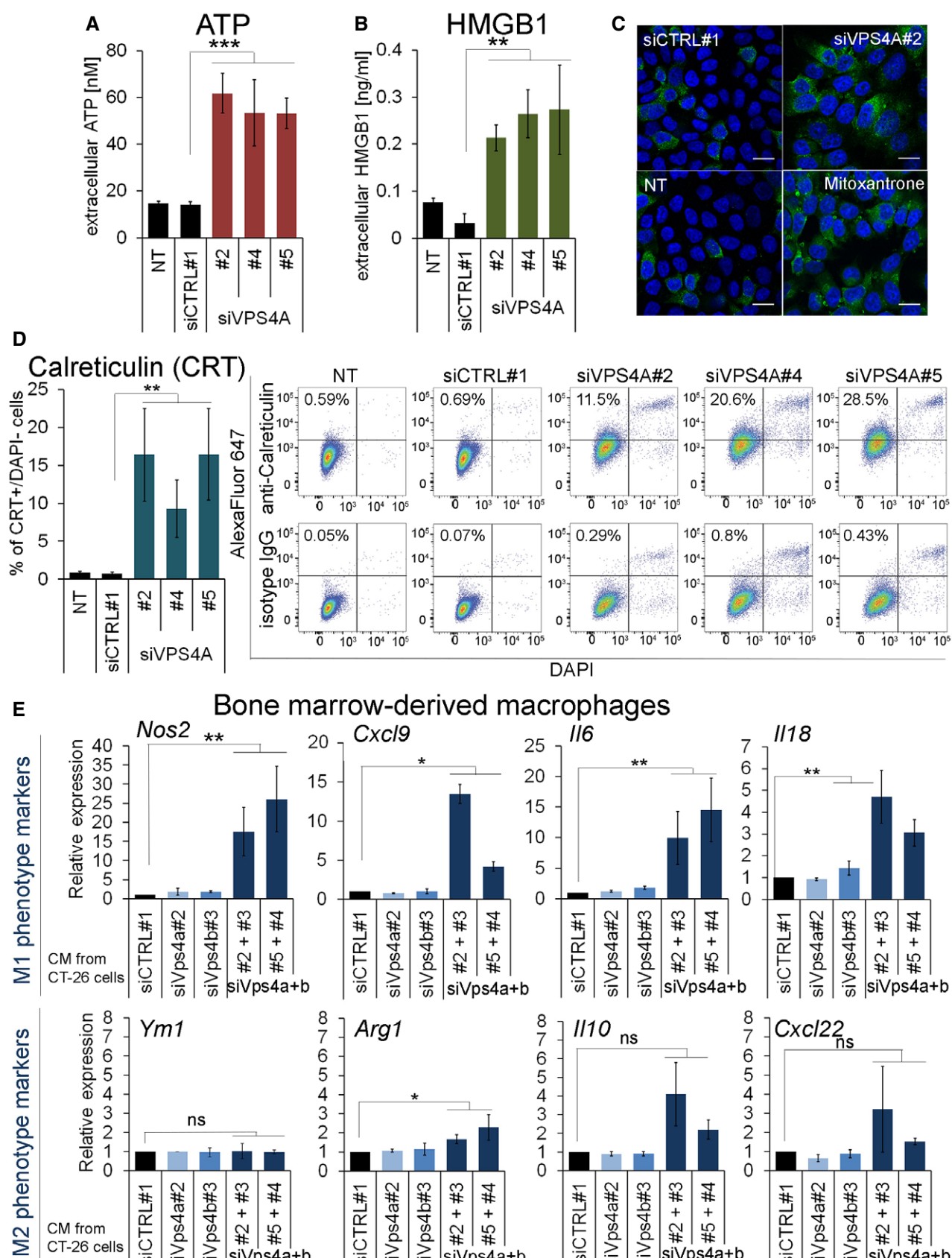

Figure 6.

**Figure 6.  Synthetic lethality between *VPS4A* and *VPS4B* induces release of immunogenic DAMPs and promotes M1 macrophage polarization.**

A, B    Measurement of ATP (A) and HMGB1 (B) released to the cell medium by HCT116 *VPS4B*$^{-/-}$ cells non-transfected (NT) or transfected with siRNA (non-targeting siCTRL#1 or targeting siVPS4A duplexes: #2, #4, or #5). Cell culture media were exchanged 16 h after transfection, and fresh media were conditioned for the next 52-58 h. For non-transfected cells (NT), the same treatment protocol was used but without the transfection mixture. Data are means of five (A) or four (B) independent experiments ± SEM. Two-tailed unpaired *t*-test; **$P < 0.01$, ***$P < 0.001$.

C    Microscopy images presenting cell surface calreticulin (green) in HCT116 *VPS4B*$^{-/-}$ cells 48 h after transfection with siRNA (non-targeting siCTRL#1 or targeting siVPS4A#2). As a positive control for detection of cell surface calreticulin, non-transfected cells (NT) were treated with 2 μM mitoxantrone for 24 h. In blue, DAPI staining. Scale bar, 15 μm.

D    Flow cytometric analysis of calreticulin exposed on the cell surface of VPS4A-depleted HCT116 *VPS4B*$^{-/-}$ cells 66 h after siRNA transfection (siVPS4A duplexes: #2, #4, or #5 were used). Non-transfected (NT) or siCTRL#1-transfected cells served as negative controls. Left panel, percentage of cells positive for calreticulin in the population of live (DAPI-negative) cells, data are means of four independent experiments ± SEM. The Mann–Whitney *U*-test; **$P < 0.01$. Right panel, representative dot plot diagrams of flow cytometric analysis of cell surface-exposed calreticulin. Primary rabbit anti-calreticulin and control isotype IgG antibodies were used for staining, followed by secondary AlexaFluor 647-conjugated antibody.

E    qPCR analysis of M1 (pro-inflammatory) and M2 (anti-inflammatory) macrophage polarization markers in mouse bone marrow-derived macrophages (BMDMs) incubated for 24 h in conditioned media (CM) collected from control (siCTRL#1), Vps4a- and/or Vps4b-depleted CT-26 cells. For double Vps4a+b depletion, various combinations of siVps4a#2 or #5 and siVps4B#3 or #4 duplexes were used. Data were normalized and are presented as the fold change of expression of a given M1 or M2 marker in BMDMs treated with CM from Vps4-depleted cells compared to its expression in BMDMs treated with CM from siCTRL#1-transfected cells (set as 1). Data are means of four independent experiments ± SEM. One-sample *t*-test; ns—non-significant ($P \geq 0.05$), *$P < 0.05$, **$P < 0.01$.

Data information: The exact *P*-values can be found in the source data for this figure.
Source data are available online for this figure.

lethality between *VPS4A* and *VPS4B* is conserved across tumor types (CRC, lung, pancreas; Fig 2) and species (Fig EV5). Combined, these features fulfill the criteria for highly penetrant synthetic lethal interactors proposed by Ryan *et al* (2018). Therefore, we predict that the dependency on *VPS4A* will occur across many VPS4B-deficient cells within the tumor mass, irrespective of their genetic background.

At present, the lack of a selective inhibitor for VPS4 raises a serious limitation to further evaluate the synthetic lethality approach in treating VPS4B-deficient CRC. The crystal structures of mammalian VPS4B and its yeast ortholog show similarities in their ATPase domains (Scott *et al*, 2005; Xiao *et al*, 2007; Hartmann *et al*, 2008; Inoue *et al*, 2008; Sun *et al*, 2017). Recent attempts to develop inhibitors for VCP/p97 (a distinct member of the type II AAA$^+$ ATPase family, overexpressed in many cancers) identified two compounds that inhibit yeast VPS4 (Zhang *et al*, 2016) and human VPS4B (Pohler *et al*, 2018). Even though the low selectivity of these molecules excludes their usage in the *VPS4A + B* synthetic lethality-based approach, it confirms the druggability of VPS4 and yields optimism for the future development of a selective VPS4 inhibitor. Although obtaining a specific VPS4A inhibitor may be infeasible (due to the high identity between paralogs), a pan-VPS4 inhibitor could still be useful, because VPS4B-deficient cancer cells are likely more sensitive to VPS4 inhibition than normal cells. These predictions imply a therapeutic window for safe dosage of such a pan-VPS4 inhibitor. Moreover, in the course of revision of this manuscript, thanks to the new dataset deposited in the DepMap portal, we verified that even a partial loss of *VPS4B* expression renders cancer cells more vulnerable to VPS4A perturbation (Fig 2). In our opinion, this suggests that a VPS4 inhibitor would exert a therapeutic effect also in tumors with an incomplete loss of *VPS4B* expression.

Most anti-cancer therapies trigger some programs of cell death (Wang *et al*, 2018; Messmer *et al*, 2019). Dying cells then emit signals that coordinate various adaptive responses within the tumor environment. Consequently, differential modulation of these signals may compromise or boost immunological control over a tumor leading to unwanted or beneficial therapeutic outcomes. Better understanding of specific cell death inducers and programs is paramount

to delineate the interconnectivity among various death pathways and their impact on immune cells for successful clinical translation (Garg & Agostinis, 2017; Messmer *et al*, 2019). Here, we reveal that concomitant depletion of VPS4A+B induces irreversible cell damage triggering two cell death pathways that may operate independently of each other (Figs 4 and 5). The first, caspase-dependent apoptosis, is reminiscent of apoptosis caused by dVps4 deficiency in Drosophila (Rodahl *et al*, 2009). The second is a caspase-independent, RIPK1-mediated process. These two death pathways were activated in VPS4A+B-depleted CRC cells that cannot undergo classical necroptosis, due to the downregulation of RIPK3 (typical for many cancer types). Based on this finding, we suggest that most VPS4B-deficient cancer cells would not avert cell death upon therapeutic VPS4A perturbation, irrespective of their individual genetic or epigenetic alterations in one or more death pathways that may appear in a heterogenic tumor mass. Thus, upregulation of anti-apoptotic factors occurring in some cancers would not make them inherently resistant to therapies targeting VPS4, therefore eliminating one mechanism of potential resistance.

The induction of cell death upon VPS4A+B depletion correlated with cell-autonomous activation of inflammatory signaling mediated by the NF-κB pathway and expression of immunomodulatory cytokines (Figs 4 and 5). At the cellular level, this inflammatory response induced upon *VPS4A+B* synthetic lethality can be viewed as an example of sterile inflammation caused by intracellular dysfunction likely of numerous membrane organelles. Similar types of responses have been recently shown for the dysfunction of the ER or mitochondria (West *et al*, 2015; Keestra-Gounder *et al*, 2016) and by us for endosomes (Maminska *et al*, 2016). In addition, VPS4A+B-depleted dying cells secreted highly immunogenic DAMPs (ATP, HMGB1, and calreticulin; Fig 6) that are well-established hallmarks of ICD (Kepp *et al*, 2014). ICD, unlike other tolerogenic cell death types, can induce an effective anti-tumor adaptive response by activating dendritic cells and subsequently specific T cells. ICD switches the tumor into an endogenous vaccine, which holds therapeutic promise (Vandenberk *et al*, 2015; Garg *et al*, 2016; Montico *et al*, 2017). Anthracyclines and some physical methods (e.g., photodynamic therapy) can induce ICD (Galluzzi *et al*, 2017). To

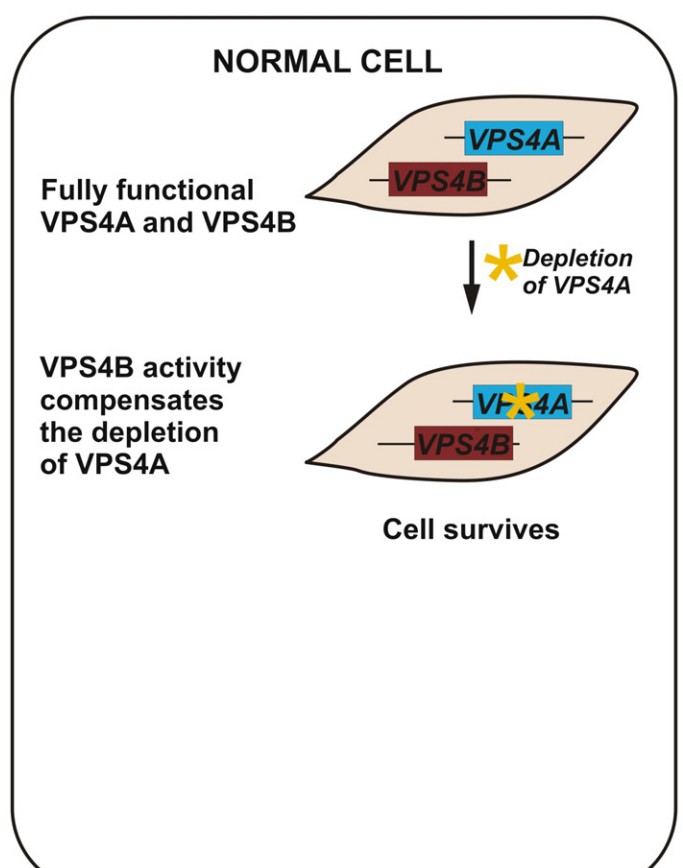
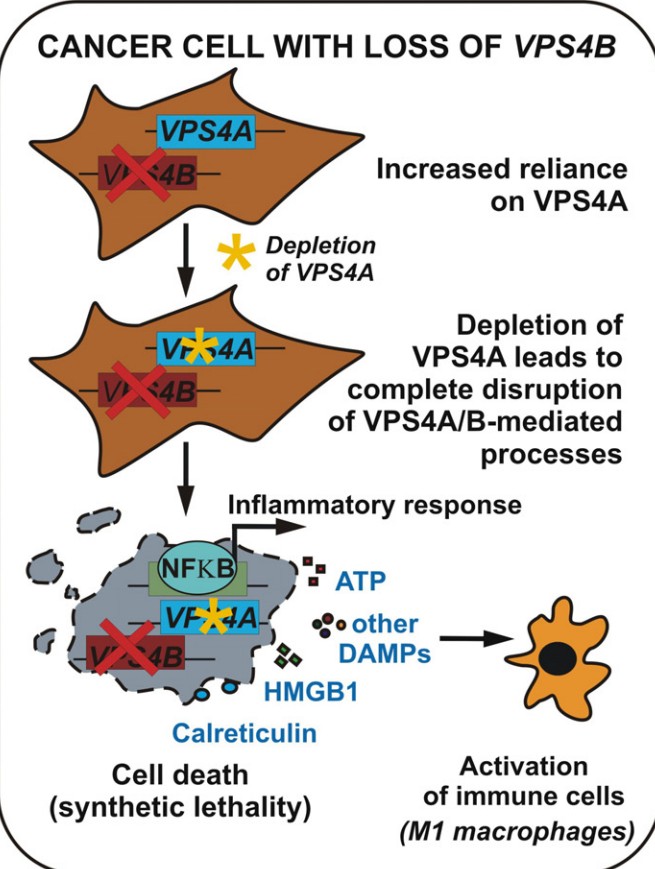

**Figure 7. Model for synthetic lethal interaction between *VPS4A* and *VPS4B*.**

Left panel, in normal cells, both VPS4A and VPS4B act redundantly in several essential intracellular processes. So, a single depletion of any VPS4 paralog (e.g., *VPS4A*) is tolerated, as unperturbed expression of the other paralog alone (e.g., *VPS4B*) suffices to substitute for its downregulated counterpart. Right panel, cells that have lost *VPS4B* expression, e.g., due to oncogenic genome rearrangements, rely exclusively on VPS4A activity. So, inactivation of VPS4A in these cells leads to synthetic lethality that is accompanied by strong induction of an inflammatory response and release of immunogenic DAMPs. Immunomodulatory molecules released by dying VPS4A+B-deficient cancer cells can elicit paracrine effects on primary immune cells, e.g., reprogramming of macrophages toward the M1 anti-tumor phenotype.

our knowledge, synthetic lethality caused by depletion of two gene products has not been shown to trigger ICD until now. Since we observed immunomodulatory DAMPs released by dying CRC cells, we propose an immunogenic nature of cell death elicited by VPS4A+B depletion. Our view is further supported by the finding that DAMP-releasing dying cells could initiate T-cell immunity only with activated RIPK1 and NF-κB (Yatim *et al*, 2015). Consistently, we showed that these two pathways operated in dying VPS4A+B-depleted CRC cells. However, future *in vivo* experiments in immunocompetent mice will ultimately confirm immunogenicity of cell death driven by *VPS4A + B* synthetic lethality.

Finally, we believe that cell death induced by VPS4A+B depletion may stimulate not only adaptive but also innate immune responses, because the conditioned medium from dying Vps4a+b-depleted mouse colon carcinoma cells induced an anti-tumor M1 phenotype in mouse macrophages (Fig 6). This initial observation may have translational potential, as reprogramming tumor-associated macrophages toward the M1 phenotype by anti-cancer therapies has been recently proposed as a promising strategy to increase effectiveness of combined treatments (Genard *et al*, 2017). In summary, our

findings establish a foundation for future work aiming to develop a VPS4 inhibitor as a putative therapeutic for precision therapy of VPS4B-deficient cancers, such as CRC. Our data further suggest that targeting the VPS4 activity in cancer cells may yield an inflammatory cell death program that favors the induction of anti-tumor innate and adaptive immune responses.

## Materials and Methods

### Cell culture

Human cell lines were obtained from the following sources: wild-type HCT116 (HD PAR-073, HCT116 $VPS4B^{+/+}$) and two CRISPR/Cas9 engineered knockout clones (HCT116 $VPS4B^{-/-}$) were supplied by Horizon Discovery Ltd (more details in a section: Generation of *VPS4B* knockout cell line); RKO (CRL-2577), DLD-1 (CCL-221), CCD-1070Sk (CRL-2091), and CCD-841CoN (CRL-1790) cell lines were obtained from American Type Culture Collection (ATCC); HOP62 and SNU410 cell lines were from the repository of National

Cancer Institute in Frederick (USA) and Korean Cell Line Bank (South Korea), respectively. SW480 and mouse colon carcinoma CT-26 cell lines were from collection of the Maria Skłodowska-Curie Institute-Oncology Centre in Warsaw. HCT116, HOP62, and CT-26 were maintained in Roswell Park Memorial Institute-1640 medium (RPMI, Sigma-Aldrich, R0883), SNU410 in RPMI with HEPES (Sigma-Aldrich, R5886), RKO and CCD-841CoN in Eagle's minimum essential medium (EMEM, ATCC, 30-2003), DLD-1 and SW480 in Dulbecco's modified Eagle's medium (DMEM, Sigma-Aldrich, D8062), and CCD-1070Sk in minimum essential medium (MEM, Sigma-Aldrich, M2279). RPMI, MEM, and DMEM were supplemented with 10% (v/v) fetal bovine serum (FBS, Sigma-Aldrich, F7524) and 2 mM L-glutamine (Sigma-Aldrich, G7513). EMEM was supplemented with 10% (v/v) FBS. During the study, cell lines were regularly tested for mycoplasma and the identities of HCT116, DLD-1, and RKO were confirmed by short tandem repeat (STR) profiling performed by the ATCC Cell Authentication Service.

**Transfection of human CRC cell lines with small interfering RNAs (siRNAs)**

Cells were forward- or reverse-transfected with siRNAs using Lipofectamine RNAiMAX transfection reagent (Thermo Fisher Scientific, 13778150) and protocols provided by the manufacturer. The concentration of single on-target siRNA duplex used for transfection was 20–40 nM. In experiments with simultaneous knockdown of two proteins, 20–40 nM of each siRNA duplexes was used (in controls 40–80 nM of non-targeting siRNA, respectively). At least two on-target siRNA duplexes were used independently to target mRNA of a given gene. The following Pre-Design or Validated Ambion Silencer Select siRNAs (Thermo Fisher Scientific) were used: non-targeting siCTRL#1 (4390843) and siCTRL#2 (4390846); on-target siVPS4A#1 (s25966), siVPS4A#2 (s25968), siVPS4A#3 (s25967), siVPS4B#1 (s18272), siVPS4B#2 (s18273), siRELA#1 (s11916), siRELA#2 (s11915). Additionally, two custom-ordered Silencer Select duplexes were used: siVPS4A#4 (sense strand 5'->3' CCACAAACAUCC CAUGGGUtt, antisense strand 5'->3' ACCCAUGGGAUGUUUGUGGtt), siVPS4A#5 (sense strand 5'->3' UCAAAGAGAACCAGAGUGATT, antisense strand 5'->3' UCACUCUGGUUCUCUUUGAtt).

**Generation of *VPS4B* knockout cell line**

*VPS4B* knockout in HCT116 cell line (HCT116 *VPS4B*$^{-/-}$) was generated using the CRISPR/Cas9 genome editing technology by Horizon Discovery Ltd. One sgRNA targeting exon 3 was used (5'->3' TGATA GAGCAGAAAAAACTAA). Bi-allelic knockout of *VPS4B* in two selected clones was verified by Sanger sequencing of the amplified VPS4B region containing sgRNA target site and by immunoblotting of cell lysates with anti-VPS4B antibody (Atlas Antibodies, HPA057649, 1:1,000; Figs EV1C4 and C5, and EV3A). A proper isogenic control cell line (HD PAR-073, HCT116 *VPS4B*$^{+/+}$) was provided by the company along with the engineered HCT116 *VPS4B*$^{-/-}$.

**Construction of HCT116 cells with inducible depletion of VPS4A by short hairpin RNA (shRNA)**

To generate HCT116 *VPS4B*$^{-/-}$ cells with doxycycline-inducible knockdown of VPS4A (HCT116 VPS4B$^{-/-}$ shVPS4A), SMARTvector

Inducible Lentiviral shRNA Vector system (Dharmacon/Horizon Discovery) was used. To select cells with the highest silencing efficiency of *VPS4A*, three different shRNA constructs (SMARTvector Inducible Human VPS4A mCMV-TurboGFP shRNA) were tested: shVPS4A#1 (V3SH11252-226989676), shVPS4A#2 (V3SH11252-226390198), and shVPS4A#3 (V3SH11252-225581302). Additionally, to generate proper control lines two separate constructs bearing inducible non-targeting shRNA sequences (SMARTvector Inducible Non-targeting mCMV-TurboGFP, shCTRL#1—VSC11651; and shCTRL#2—VSC11495) were used. All procedures were performed according to the manufacturer's instructions. In brief, lentiviruses were prepared by transfecting HEK293 cells using Trans-Lentiviral shRNA Packaging System (Dharmacon/Horizon Discovery, TLP5913). Virus-containing supernatants were harvested 48 h after transfection and used to infect HCT116 *VPS4B*$^{-/-}$. Transduced cells were selected with 1 µg/ml puromycin (Toku-E, P001) for 3 days. To induce shRNA expression, puromycin-resistant cells were cultured for 3 days in medium supplemented with 1 µg/ml doxycycline (MP Biomedicals, 0219895501). Cell lysates were analyzed by immunoblotting with anti-VPS4 antibody (Santa Cruz Biotechnology, sc-1333122,1:1,000) to confirm doxycycline-induced VPS4A protein depletion.

**Cell viability and BrdU incorporation assays**

$3 \times 10^3$ of HCT116 or RKO, $4 \times 10^3$ of SW480, $2.9 \times 10^3$ of DLD-1, $1.4 \times 10^3$ of SNU410, and $1.2 \times 10^3$ of HOP62 cells were reverse-transfected with siRNA in 96-well plates (cell numbers are given per well). Viability of HCT116, RKO, and SW480 cells was measured 96 h post-transfection, of HOP62—after 144 h, of SNU410—after 168 h using ATPlite test (PerkinElmer, 6016947) according to the manufacturer's protocol. BrdU Cell Proliferation ELISA assay (Roche, 11647229001) was used to assess the growth of DLD-1 cells. The assay was performed according to the manufacturer's instructions with the following modifications. BrdU reagent was added 5 h prior to cell fixation. For detection, 100 µl of substrate solution was added for 5 min followed by addition of 25 µl 1 M $H_2SO_4$. The colorimetric signal was detected at 450 nm. Viability measurements of necrostatin-1- or Q-VD-Oph-treated cells were performed as follows: 24 h after siRNA transfection of HCT116 *VPS4B*$^{-/-}$ cells, 50 µM necrostatin-1, 20 µM Q-VD-Oph, or vehicle (DMSO) was added to the cell medium for 48 h. Cell viability was measured using Cell Counting Kit-8 (Dojindo Molecular Technologies, CK04) according to the manufacturer's protocol. For the assessment of growth of HCT116 *VPS4B*$^{-/-}$ cells bearing doxycycline-induced shRNA expression (shVPS4A#1-2 or shCTRL#1-2), $1 \times 10^4$ cells were seeded per well of 96-well plate in doxycycline (1 µg/ml) supplemented cell medium. Cell growth was analyzed each day for 5 consecutive days using Cell Counting Kit-8.

**Clonogenic assay**

Cells (non-transfected or 24 h after siRNA transfection) were seeded at the density of 400 or 2,000 per well of 6-well plates and grown for 14 days to form colonies. For staining, colonies were washed with PBS, fixed for 5 min in acetic acid:methanol solution 3:1 (v/v), and next incubated for 10 min in 0.2% crystal violet solution in 70% ethanol for 10 min. Plates with colonies were scanned using

Odyssey Infrared Imaging System (LI-COR, Biosciences), and images were analyzed as previously described (Guzman *et al*, 2014).

## Antibodies

The primary antibodies used for immunoblotting were as follows: rabbit anti-VPS4B (HPA057649, 1:1,000) from Atlas Antibodies; mouse anti-VPS4A (sc-133122, 1:500 – 1:1,000), mouse anti-caspase 8 (sc-56070, 1:1,000), mouse anti-PARP (sc-8007, 1:1,000), rabbit anti-GAPDH (sc-25778; 1:1,000) from Santa Cruz Biotechnology; rabbit anti-VPS4A (SAB4200022, 1:500), mouse anti-vinculin (V9131; 1:1,000) from Sigma-Aldrich; rabbit anti-caspase 3 (9665, 1:1,000), rabbit anti-caspase 7 (12827, 1:1,000), rabbit anti-cleaved caspase 7 (8438, 1:1,000), rabbit anti-cleaved-caspase 3 (9664, 1:1,000), rabbit anti-cleaved caspase 9 (7237, 1:1,000), rabbit anti-phospho-RelA (3033, 1:1,000), rabbit anti-p100/p52 (4882, 1:1,000), mouse anti-phospho-IκBα (9246S, 1:1,000), mouse anti-IκBα (4814S, 1:1,000) from Cell Signaling Technology; rabbit anti-RelA (14-6731-81, 1:1,000) from EBioscience; mouse anti-p100/p52 (05-361, 1:1,000) from Millipore. Secondary horseradish peroxidase-conjugated anti-mouse (111-035-062, 1:10,000) and anti-rabbit (111-035-144, 1:10,000) antibodies were from Jackson ImmunoResearch.

The primary antibodies used for immunofluorescence were as follows: rabbit anti-calreticulin (ab2907, 1:200) from Abcam, mouse anti-EEA1 (610457, 1:200) from BD Biosciences, rabbit anti-Rab7 (R4779, 1:100) from Sigma-Aldrich, and rabbit anti-LAMP1 (cs9091, 1:400) from Cell Signaling. The secondary antibodies were as follows: donkey AlexaFluor 647-conjugated anti-rabbit (A31572, 1:400) from Thermo Fisher Scientific and goat AlexaFluor 488-conjugated anti-mouse and anti-rabbit (A11029, 1:400 and A11034, 1:400, respectively) from Life Technologies.

The antibodies used for immunohistochemistry were as follows: rabbit anti-VPS4B (HPA057649, 1:100) from Atlas Antibodies and mouse anti-VPS4 (recognizing only VPS4A, Fig EV2B, sc-122133, 1:50) from Santa Cruz Biotechnology.

The antibodies used for flow cytometry were as follows: rabbit anti-calreticulin (ab2907, 1:200), rabbit IgG (ab171890, 1:200) from Abcam, and donkey AlexaFluor 647-conjugated anti-rabbit (A31572, 1:350) from Thermo Fisher Scientific.

## Immunoblotting and densitometry analysis

Cells were lysed in RIPA buffer (1% Triton X-100, 0.5% sodium deoxycholate, 0.1% SDS, 50 mM Tris (pH 7.4), 150 mM NaCl, 0.5 mM EDTA) supplemented with protease inhibitor cocktail (6 μg/ml chymostatin, 0.5 μg/ml leupeptin, 10 μg/ml antipain, 2 μg/ml aprotinin, 0.7 μg/ml pepstatin A, and 10 μg/ml 4-amidinophenylmethanesulfonyl fluoride hydrochloride; Sigma-Aldrich) and phosphatase inhibitor cocktails (Sigma-Aldrich, P0044 and P5726). Protein concentration was measured with BCA Protein Assay Kit (Thermo Fisher Scientific, 23225). 30–50 μg of total protein per sample was resolved on 12% or 14% SDS–PAGE, transferred to nitrocellulose membrane (Amersham Hybond, GE Healthcare Life Science, 10600002), probed with specific primary and secondary antibodies, and developed using the detection solution (Bio-Rad, 170-5061) and ChemiDoc imaging system (Bio-Rad). Densitometry of protein bands was carried out using ImageJ software (Schneider *et al*, 2012). GAPDH or vinculin bands were used as internal loading controls.

## Transcriptome analysis by RNA sequencing

Sequencing libraries were generated using Ion AmpliSeq Transcriptome Human Gene Expression Panel (Thermo Fisher Scientific). Sequencing was performed using Ion Proton instrument with 7 or 8 samples per chip with Ion PI Hi-Q Sequencing 200 Kit (Thermo Fisher Scientific). Reads were aligned to the hg19 AmpliSeq Transcriptome ERCC v1 with Torrent Mapping Alignment Program (version 5.0.4, Thermo Fisher Scientific). Transcripts were quantified with HTseq-count (version 0.6.0) run with default options (Anders *et al*, 2015).

Gene-level differential expression analysis was performed with the aid of the R package DESeq2 (version 1.18.1; Love *et al*, 2014) for genes with at least 10 reads across conditions and by taking into account the batch effect and applying the following contrasts ($\alpha = 0.05$): NT (non-transfected) versus siCTRL#1 (non-targeting control siRNA#1), NT (non-transfected) versus siCTRL#2 (non-targeting control siRNA#2), NT versus siVPS4A#1, NT versus siVPS4A#2, NT versus siVPS4B#1, NT versus siVPS4B#2, NT versus siVPS4A#2 + siVPS4B#1 (referred thereafter as siVPS4A+B#1), NT versus siVPS4A#1 + siVPS4B#2 (referred thereafter as siVPS4A+B#2), siCTRL#1 versus siCTRL#2, siCTRL#1 versus siVPS4A#1, siCTRL#1 versus siVPS4A#2, siCTRL#1 versus siVPS4B#1, siCTRL#1 versus siVPS4B#2, siCTRL#1 versus siVPS4A+B#1, siCTRL#1 versus siVPS4A+B#2, siCTRL#2 versus siVPS4A#1, siCTRL#2 versus siVPS4A#2, siCTRL#2 versus siVPS4B#1, siCTRL#2 versus siVPS4B#2, siCTRL#2 versus siVPS4A+B#1, siCTRL#2 versus siVPS4A+B#2. Differentially expressed genes were combined into a single list excluding non-protein-coding genes. While running exploratory data analysis (namely, principal component analysis and heatmap of sample-to-sample distances), the NT and siCTRL#1 control conditions displayed consistent behavior, and therefore, they were used for downstream analysis.

The overlap for different silencing conditions and normalization contrasts was visualized using the VennDiagram package (version 1.6.20). The genes, which overlapped for pairs of on-target siRNAs normalized against either NT or siCTRL#1-transfected patterns, were subjected to GO analysis of biological processes and Reactome pathway analysis using clusterProfiler (version 3.6.0; Yu *et al*, 2012) and ReactomePA R-packages (version 3.8; Yu & He, 2016) taking advantage of enrichGO and enrichPathway functions, respectively. All enrichment p-values in GO analysis were corrected for multiple testing using the Benjamini–Hochberg method, and only genes with adjusted *P*-value < 0.05 were considered significant. The minimal and maximal sizes of gene clusters were set to 10 and 500, respectively. Redundant terms were removed by means of the simplify function with cutoff 0.6. Gene set enrichment analysis (GSEA) was executed using the clusterProfiler function gseGO with 1000 permutations, Benjamini–Hochberg correction for multiple testing and the sizes of gene clusters were in range of 10–500. Hierarchical clustering of the genes associated with selected Gene Ontology terms was performed on variance-stabilizing transformed data using Euclidean distances and complete algorithm. Heatmaps were plotted using pheatmap (version 1.0.10, Raivo Kolde (2018); pheatmap: Pretty Heatmaps). All calculations were performed in R version 3.4.4 (https://www.R-project.org).

## Quantitative real-time PCR (qRT–PCR)

Total RNA was isolated with High Pure Isolation Kit (Roche, 11828665001). For cDNA synthesis random nonamers, oligo(dT) 23 and M-MLV reverse transcriptase (Sigma-Aldrich, R7647, O4387, and M1302, respectively) were used according to manufacturer's instructions. Expression of genes of interest was measured using primers designed with the NCBI tool (and custom-synthesized by Sigma-Aldrich) listed in Appendix Table S2 or the following TaqMan® Gene Expression Assays: Hs99999903_m1 for *ACTB*, Hs00203085_m1 for *VPS4A* and Hs00191617_m1 for *VPS4B* (Thermo Fisher Scientific). The qRT–PCR mixture was performed with the Kapa Sybr Fast qPCR Kit (KapaBiosystems, KK4618) or TaqMan® Gene Expression Master Mix (Thermo Fisher Scientific, 4369016) using a 7900HT Fast Real-Time PCR thermocycler (Applied Biosystems) with two technical repeats per experimental condition. The data were normalized according to the level of housekeeping genes *ACTB* or *Rpl19* and presented as fold changes.

## Copy number analysis with qPCR

Target (*VPS4B*; assay ID:Hs03033551_cn, *BCL2*; assay ID: Hs01601779_cn) and reference probes (RNaseP—4403326) were selected using the assay search tool on the Thermo Fisher Scientific website. All reactions with TaqMan Copy Number Assays were performed in parallel using the FAM dye label-based assay for the target of interest and the VIC dye label reference assay. Amplification reactions (5 μl), which were performed in triplicate, consisted of: 40 ng genomic DNA, 1X TaqMan Copy Number Assay, 1X TaqMan Copy Number Reference Assay, RNase P, 1X SensiFAST Probe Hi-ROX kit (BIO-82020, Bioline). PCR was performed with an Applied Biosystems HT7900 Real-Time PCR system using the default universal cycling conditions starting with 95°C for 10 min followed by 40 cycles: 95°C for 15 s, 60°C for 60 s. Data were analyzed with SDS v2.4.1 software (Applied Biosystems). Result export files were opened in CopyCaller™ Software v2.0 for sample copy number analysis by the relative quantitation method.

## Extracellular ATP and HMGB1 assays

$1 \times 10^5$ of HCT116 *VPS4B*$^{-/-}$ cells were seeded per well of 12-well plates and either left non-transfected or forward-transfected with siRNAs (control or targeting *VPS4A*). After 16-h growth, media were removed and cells were cultured in 0.6 ml of fresh medium for further 52–56 h. Next, media were collected and centrifuged at $300 \times g$ for 10 min, followed by centrifugation at $2,000 \times g$ for 10 min. Media stored at $-80°C$ were used for the following measurements: ATP was quantified using ENLITEN® ATP Assay System (Promega, FF2000) and HMGB1 was quantified using HMGB1 ELISA Kit (Aviva Systems Biology, OKCD04074). All procedures were performed according to the manufacturers' instructions.

## Flow cytometric analysis

HCT116 *VPS4B*$^{-/-}$ cells ($2 \times 10^5$ per well of 6-well plates) were forward-transfected with siRNAs or left non-transfected. For cell surface calreticulin staining, 66 h after transfection cells were harvested with trypsin and centrifuged for 5 min at $200 \times g$. Cells were blocked (3% FBS in PBS, 10 min, 4°C) and incubated with rabbit anti-calreticulin antibody (Abcam, ab2097, 1:200, 30 min, 4°C) or rabbit isotype control IgG (Abcam, ab171870, 1:200, 30 min, 4°C) followed by AlexaFluor 647-conjugated secondary antibody (Thermo Fisher Scientific, A31572, 1:350, 30 min, 4°C). Cell aggregates were removed from analysis by doublet discrimination using FSC-A versus FSC-W and SSC-A versus SSC-W parameters. To discriminate live and dead cells, DAPI (0.5 μg/ml) was added 5 min before flow cytometry analysis.

For transferrin uptake analysis, 72 h after transfection Alexa-Fluor 647-conjugated transferrin (T23366, Thermo Fisher Scientific) was administered to the cell medium (final concentration 25 μg/ml) for 10 min. Then, cells were washed, harvested with trypsin, and washed twice with ice-cold PBS. After fixing for 15 min in 3.5% paraformaldehyde, cells were washed twice with PBS and analyzed.

For cell cycle analysis, 72 h after siRNA transfection cells were harvested, washed twice with PBS, and fixed for 1 h in ice-cold 70% ETOH. Washed cells were then incubated in extraction buffer (4 mM citric acid in 0.2 M Na$_2$HPO$_4$, 5 min at room temperature) followed by staining buffer (3.8 mM sodium citrate, 50 μg/ml propidium iodide, and 0.5 mg/ml RNase A, 30 min at room temperature).

All analyses were performed using BD LSRFortessa flow cytometer (BD Biosciences). A total of 10,000–50,000 cells were counted for each treatment condition. Flow cytometry data were plotted and analyzed by FlowJo (Tree Star Inc.) and ModFit LT (Verity Software House) software.

## Immunofluorescence staining

$4.5 \times 10^4$ of HCT116 cells were seeded on fibronectin or poly-L-lysine (Merck)-coated coverslips, next day forward-transfected with control or *VPS4*-targeting siRNA, incubated for 48–52 h, and stained using one of the following protocols. For cell surface calreticulin staining, a positive control for detection of cell surface calreticulin was HCT116 *VPS4B*$^{-/-}$ cells treated for 24 h with 2 μM mitoxantrone (Abcam, ab141041). Transfected or mitoxantrone-treated cells were then fixed for 15 min with 3% paraformaldehyde and blocked with 2% albumin in PBS for 15 min. Next, cells were incubated for 1 h with anti-calreticulin antibody (Abcam, ab2097, 1:200) followed by incubation with secondary antibody (Thermo Fisher Scientific, A31572, 1:400). For staining of endocytic proteins, after fixation and blocking cells were permeabilized with 0.1% saponin, incubated for 2 h with primary antibodies, followed by 1-h incubation with secondary antibodies. DAPI or Hoechst 33342 were used for nuclei staining. Coverslips were mounted in Mowiol (Sigma-Aldrich) or the ProLong Diamond Antifade Mountant (Thermo Fisher Scientific) and imaged using Zeiss LSM710 or Zeiss LSM800 confocal microscopes with 40×/1.30 or 60×/1.40 oil immersion objectives and ZEN 2009 software. Images were processed in ImageJ software with only linear brightness/contrast corrections. Quantitative analysis of endocytic vesicles was performed in ImageJ software. 3D Object Counter plug-in (Bolte & Cordelieres, 2006) was used to determine the intensities and sizes of the vesicles for $n > 4$ z-stacks (ca. 30 cells per condition). The high-intensity EEA1-positive vesicles were identified manually in the images, and the same intensity-threshold was used for all the conditions. Results are

presented as a percentage of high-intensity vesicles to all counted vesicles in each $z$-stack.

## Transfection of mouse CT-26 cells and collection of cell medium

$1.5 \times 10^5$ of CT-26 cells seeded per well of 12-well plate were reverse-transfected with siRNA. The following Pre-Design or custom-ordered Ambion Silencer Select siRNAs (Thermo Fisher Scientific) were used: non-targeting siCTRL#1 (4390843); siVPS4A#2 (also named siVps4a#2, since it targets *VPS4A* in human and mouse, s25968), siVPS4A#5 (also named siVps4a#5, since it targets *VPS4A* in human and mouse, sense strand 5′->3′ UCAAAGAGAACCAGA GUGATT, antisense strand 5′->3′ UCACUCUGGUUCUCUUUGAtt), siVps4b#3 (s231857) and siVps4b#4 (s231858). In control samples, the final concentration of non-targeting siRNA was 60 nM; 30 nM on-target siRNA + 30 nM of control siRNA were used for single silencing of *Vps4a* or *Vps4b*; 30 nM of each targeting siRNA was used for simultaneous silencing of *Vps4a + b*. 16 h after transfection, cells were washed with fresh medium. Conditioned cell media were collected 72 h after transfection, centrifuged for 5 min at 200 $g$, and stored at −20°C. Cells were lysed and used for further immunoblotting or qPCR analysis to confirm Vps4a/b protein or mRNA knockdown efficiency (Fig EV5A and B).

## Isolation and treatment of bone marrow-derived macrophages (BMDMs)

Bone marrow stem cells were isolated from 8- to 12-week-old female C57BL/6 mice by flushing the bone marrow with PBS. $4.5 \times 10^5$ of bone marrow stem cells were seeded in a single well of non-tissue culture-treated 12-well plate and differentiated into macrophages by culturing them for 7 days in RPMI medium (Sigma-Aldrich, R2405) supplemented with 10% FBS (GE Healthcare, Hyclone, SV30160.03HI) and 10 ng/ml M-CSF (PeproTech 315-02) with fresh medium added after 4 days. To activate the M1 or M2 phenotype, differentiated BMDMs were incubated for 24 h in conditioned media (CM). CM was prepared by mixing medium collected from CT-26 cells (details above) with RPMI medium containing 10% FBS at the ratio 3:1. CM was supplemented with M-CSF (10 ng/ml). Cell lysates were processed for RNA isolation and qPCR analysis of expression levels of M1/M2 markers.

## Xenografts

All animal work was performed in accordance with the protocol approved by the 2$^{nd}$ Local Ethics Committee for Animal Experimentation in Warsaw (decision no. WAW2/047/2018). The NU/J (nude) athymic mice were purchased from the Jackson Laboratory and maintained in a specific pathogen-free (SPF) facility. Mice were kept under 12 light/12 dark cycle and housed in individually ventilated cages (Tecniplast). Randomly selected groups of males and females were used. All mice were over 6 weeks of age. For xenograft experiments, the mice were inoculated subcutaneously with $5 \times 10^6$ cells (parental HCT116 $VPS4B^{+/+}$ or HCT116 $VPS4B^{-/-}$). Xenograft growth was monitored for 19 days, and then, the mice were sacrificed. In experiments with inducible expression of shRNA, 2 weeks after subcutaneous inoculation of HCT116 cells ($VPS4B^{-/-}$ shVPS4A#1 or $VPS4B^{-/-}$ shCTRL#1) animals were divided into two groups. One

### The paper explained

#### Problem
Personalized therapies can improve the outcome of cancer patients. To develop such therapies, novel targets must be identified for selective killing of genetically diverse tumor cells. For example, a gene X cooperates with another gene Y to support the cell's functions in healthy conditions, but partner Y is lost in a cancer cell. Then, the single remaining gene X becomes an "Achilles heel" for the cancer cell since its perturbation will cause cell death. This phenomenon is called synthetic lethality and represents a promising approach for personalized oncology. More pairs of synthetically lethal genes still need to be identified.

#### Results
Here, we showed that the *VPS4B* gene was frequently deleted in many cancer types, including in colorectal cancer, which was reflected by low *VPS4B* mRNA and protein levels in colorectal cancer samples from patients. We further identified the *VPS4A* gene as a synthetic lethal partner for *VPS4B*. We demonstrated that the perturbation of VPS4A protein in a tumor cell with loss or low level of VPS4B induced the death of cells grown *in vitro* and in mice xenografted tumors. Moreover, our study revealed that upon concomitant depletion of VPS4A and VPS4B proteins, dying cancer cells secreted immunomodulatory molecules that mediated inflammatory and anti-tumor responses.

#### Impact
Our results identify a novel pair of druggable targets for personalized oncology and provide a rationale to develop VPS4 inhibitors for precision therapy of VPS4B-deficient cancers.

group of the tumor-bearing mice was given doxycycline-containing water, to induce shRNA expression. The xenograft growth was further monitored for 11 days. Tumor volumes were measured with the Peira TM900 handheld imaging device. In all experiments, a piece (~50 mg) of each tumor sample was frozen separately for further analysis. Randomly selected samples were used for subsequent immunoblotting evaluation of VPS4A knockdown efficiency (using anti-VPS4A antibody, Sigma-Aldrich, SAB4200022) or IHC staining of VPS4B (Atlas Antibodies, HPA057649, 1:100, Fig EV1C4 and C5).

## TCGA data analysis

TCGA Pan-Cancer and TCGA Colorectal Cancer (COADREAD) data were retrieved using Xena browser (https://www.biorxiv.org/content/10.1101/326470v3) and cBioPortal (Gao *et al*, 2013), respectively.

## Immunohistochemistry (IHC) and analyses of human CRC samples

The study protocol for analysis of protein levels of VPS4B and VPS4A in human normal colon and CRC samples was approved by the Bioethics Committee of the Maria Skłodowska-Curie Institute-Oncology Centre in Warsaw (decision no. 40/2017). The informed consent was obtained from all subjects. The experiments conformed to the principles set out in the WMA Declaration of Helsinki and the Department of Health and Human Services Belmont Report. High-density tissue microarrays were constructed from formalin-fixed, paraffin-embedded diagnostic samples of 100 pairs of treatment-naïve CRC tissues, and matched normal colon samples from the collection of the Maria Skłodowska-Curie

Institute-Oncology Centre. IHC was performed using automated immunohistochemical stainer (Dako Denmark A/S) and anti-VPS4B or anti-VPS4A antibodies (details above). The EnVision Detection System (Agilent) was used for detection. Samples were reviewed for abundance of VPS4 proteins in normal and neoplastic tissues by two pathologists, who were blinded to outcome. A semi-quantitative method was applied for IHC evaluation, involving a scoring system based on the staining intensity: 0—no staining; 1—weak, 2—intermediate, and 3—strong staining; staining homogeneity was above 90%.

### qPCR analysis of mRNA level of *VPS4* paralogs in human samples

Samples of adenocarcinoma ($n = 26$), adenoma ($n = 42$), and normal colon had been collected for the purpose of the previous studies (Skrzypczak *et al*, 2010; Mikula *et al*, 2011). To determine *VPS4A* or *VPS4B* transcript abundance, qRT–PCR method with SYBR Green chemistry was applied, as described previously (Skrzypczak *et al*, 2010; Mikula *et al*, 2011). The sequences of primers for *VPS4A* and *VPS4B* (named *VPS4B-2*) are listed in Appendix Table S2.

### Statistical analysis

At least three independent experiments were performed in each case. Statistical testing was performed using GraphPad Prism 5 or R (version 3.6.0) software. Dunn's test was performed with PMCMR_4.3 package. Data were analyzed for Gaussian distribution with the Kolmogorov–Smirnov test with the Dallal–Wilkinson–Lillie test for corrected *P*-value. In case of Gaussian distribution, the following parametric two-tailed tests were used: one-sample *t*-test, unpaired *t*-test, Welch *t*-test, as appropriate. In case of non-Gaussian distribution, non-parametric Wilcoxon signed rank test, Mann–Whitney *U*-test, or Kruskal–Wallis test (followed by Dunn's multiple comparison *post hoc*) were used. The significance of mean comparison is annotated as follows: ns, non-significant ($P \geq 0.05$), *$P < 0.05$, **$P < 0.01$, ***$P < 0.001$, and ****$P < 0.0001$. Exact *P*-values are provided in the source data for each main figure or in Appendix Table S3 for EV figures. No statistical methods were used to predetermine sample size.

## Data availability

The RNA-Seq datasets have been deposited to GEO under the accession number GSE128070 (https://www.ncbi.nlm.nih.gov/geo/query/acc.cgi?acc=GSE128070).

**Expanded View** for this article is available online.

## Author contributions

The research was conceived by ES, MMią and MMik. Funding was acquired by ES and MMią. Experiments were designed and performed mostly by ES with support from PN and crucial help from KK and KG (RNA-Seq), MMik (RNA-Seq and TCGA data analysis), ED-W, AS-C and MP-S (immunohistochemistry), MC and AG (animal work), and MB-O and KP (flow cytometry). The manuscript was written by ES and MMią with input from KK and MMik. Figures were assembled by ES with help of MMik, KK, and KG. ES and MMią supervised the work. All authors approved the manuscript.

## Acknowledgements

We are grateful to A. Paziewska and A. Dąbrowska for their technical support in RNA-Seq analysis. We thank P. Ślusarczyk and K. Mleczko-Sanecka for their help with BMDM isolation. We also thank M. Banach-Orłowska, M. Kaczmarek, A. Poświata, and J. Cendrowski for critical reading of the manuscript. This work was supported by Sonata grant (2016/21/D/NZ3/00637) from the National Science Center to E. Szymańska. M. Miączyńska and K. Kolmus were supported by TEAM grant (POIR.04.04.00-00-20CE/16-00), and K. Piwocka was supported by TEAM-TECH Core Facility Plus/2017-2/2 grant (POIR.04.04.00-00-23C2/17-00)—both grants from the Foundation for Polish Science co-financed by the European Union under the European Regional Development Fund. We thank Life Science Editors for editorial assistance.

## Conflict of interest

The authors declare that they have no conflict of interest.

## For more information

(i) http://www.cbioportal.org/
(ii) https://depmap.org/portal/
(iii) https://www.proteinatlas.org/

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
