## [Review Process File · EMBO Molecular Medicine]

Synthetic lethality between VPS4A and VPS4B triggers an inflammatory response in colorectal cancer

Ewelina Szymańska, Paulina Nowak, Krzysztof Kolmus, Magdalena Cybulska, Krzysztof Goryca2, Edyta Derezińska-Wołek, Anna Szumera-Ciećkiewicz, Marta Brewińska-Olchowik, Aleksandra Grochowska, Katarzyna Piwocka, Monika Prochorec-Sobieszek, Michał Mikula, Marta Miączyńska

Review timeline:

Submission date:	26 April 2019
Editorial Decision:	15 May 2019
Revision received:	17 October 2019
Editorial Decision:	6 November 2019
Revision received:	4 December 2019
Accepted:	11 December 2019

Editor: Lise Roth

Transaction Report:

1st Editorial Decision

15 May 2019

Thank you for the submission of your manuscript to EMBO Molecular Medicine. We have now heard back from the two referees whom we asked to evaluate your manuscript.

As you will see from the reports below, both referees mention the interest of the study. However, they also raise substantial concerns on your work, which should be convincingly addressed in a major revision of the present manuscript. In particular, both referees insist on including the effects of VPS4A inhibition in cancer cell lines, and on further increasing the clinical relevance of the manuscript by testing VPS4B loss/partial loss and improving the discussion.

Addressing the reviewers' concerns in full will be necessary for further considering the manuscript in our journal, and acceptance of the manuscript will entail a second round of review. EMBO Molecular Medicine encourages a single round of revision only and therefore, acceptance or rejection of the manuscript will depend on the completeness of your responses included in the next, final version of the manuscript. For this reason, and to save you from any frustrations in the end, I would strongly advise against returning an incomplete revision.

I look forward to receiving your revised manuscript.

***** Reviewer's comments *****

Referee #1 (Comments on Novelty/Model System for Author):

The experiments are performed well and support the conclusions drawn in the manuscript. However, lethality induced by combined inhibition of two paralogs is not novel and has been observed before for other pairs of paralogs. Although the authors suggest that a large fraction of colorectal cancers have lost or reduced expression of VPS4B, they do not provide any examples of such CRC cell lines and their dependency on VPS4A. In addition, they do not test the inhibition of VPS4A and B in normal cells. As such it is difficult to conclude that VPS4B low or null CRC tumors are specifically

lethal to VPS4A inhibition. This is even more concerning when an VPS inhibitor is used that inhibits both VPS4A and B proteins.

Referee #1 (Remarks for Author):

The consequence of depletion of VPS4A and B on cell proliferation and survival is interesting and well documented. However, the claim that this supports the increased sensitivity of VPS4B low or null CRC is not substantiated due to the absence of experiments in CRC tumor cell lines with low or no VPS4B expression. If the authors would like to extrapolate the findings of combined knockdown, these experiments should also be performed in different normal cell lines, preferable in vivo to show the reduced of absence of effects. This is particular important when no specific inhibitors can be developed.

It is surprising that the data in the DRIVE database indicate that SK-CO-1 cells are the most sensitive cell line to VPS4A as well as VPS4B depletion. In addition, one can observe a correlation between the viability scores in the cell line panel for VPS4A and B.

One other point is the potential correlation between proliferation and VPS4B expression. Is the expression of VPS4B affected by proliferation rate of cell cycle progression?

Referee #2 (Comments on Novelty/Model System for Author):

The idea of targeting passenger mutations is still quite novel and interesting which is why I have rated technical and novelty highly. They seem to have done a good job identifying a synthetic lethal pair that could impact a subset of CRC patients but the medical impact is unknown at this point. It is probably a long way from clinical implementation but its a good idea.

Referee #2 (Remarks for Author):

In this manuscript the authors exploit coincidental loss of VPS4B in colorectal cancer as a biomarker to predict and test VPS4A depletion as a synthetic lethal strategy for this subset of cancers. Loss of the two VPS4 paralogues caused synthetic lethality in cell line and xenograft models. The manuscript characterizes the observed synergy with respect to gene expression changes, cell death mechanisms, and the potential to elicit an inflammatory response. Overall, the promise of targeting presumed 'passenger' mutations like VPS4B has been underexploited and this study represents an interesting and important addition to this field. The manuscript is quite thorough, the data quality look good, and the work tests important mechanisms of cell killing and immune stimulation. I have some suggestions to improve the manuscript below.

On Page 5, the authors propose to use SK-CO-1 cells as a model for a cell line with native loss of VPS4B but then find that the model is not appropriate for this purpose? Why do they include this section at all? It would be better to remove it, or screen other cell lines with natively low or lost VPS4B.

The rationale for looking at gene expression changes by RNA-seq is not clear because VPS4-depleted cells would be expected to have defects in secretion or the endomembrane system. The authors seem to observe a gene expression program consistent with dying cells, which is good, but does not add much to the mechanism of lethality. Did the authors look directly at secretory defects in the double-depleted VPS4A/4B cells?

The finding that both caspase-dependent and independent pathways lead to cell killing in the VPS4A/B depletion model is good but I thought could be framed in the text as eliminating potential resistance mechanisms (i.e. upregulation of anti-apoptotic factors seen in some cancers would NOT make them inherently resistant to this approach).

It was not clear in what genetic contexts the authors think this approach would be effective. The authors claim that 70% of CRCs have loss of VPS4B, but that only 2% have biallelic loss. Their model systems have a complete VPS4B loss, so would the synthetic lethality only be effective in the 2% with biallelic loss, or would there be some efficacy of VPS4A knockdown in the 70%? I felt that the manuscript should be explicit about this, and perhaps test partial VPS4B loss for the efficacy of

VPS4A-knockdown mediated killing. The future clinical impact and use of VPS4B as a biomarker depends on a clear statement here.

Minor points:

On page 10, the authors state that VPS4A and 4B are functionally redundant. I recommend saying 'partly' or 'partially' redundant since their own analysis (e.g. RNA-seq) shows they are not full redundant.

The grey inset images in Figure 2 are not very useful or easy to see.

Figure 2. What is the rationale for using a one-sample T-test?

Figure 3. I did not understand why the authors showed the right panel on Figure 3b. Are these just the endpoint values? Why is there no indication of statistical significance?

1st Revision - authors' response

17 October 2019

Referee #1

The experiments are performed well and support the conclusions drawn in the manuscript. However, lethality induced by combined inhibition of two paralogs is not novel and has been observed before for other pairs of paralogs. Although the authors suggest that a large fraction of colorectal cancers have lost or reduced expression of VPS4B, they do not provide any examples of such CRC cell lines and their dependency on VPS4A. In addition, they do not test the inhibition of VPS4A and B in normal cells. As such it is difficult to conclude that VPS4B low or null CRC tumors are specifically lethal to VPS4A inhibition. This is even more concerning when an VPS inhibitor is used that inhibits both VPS4A and B proteins.

Referee #1 Remarks for Author

1. The consequence of depletion of VPS4A and B on cell proliferation and survival is interesting and well documented. However, the claim that this supports the increased sensitivity of VPS4B low or null CRC is not substantiated due to the absence of experiments in CRC tumor cell lines with low or no VPS4B expression (*issue raised also by Referee #2*).

We do agree with the opinion of both Reviewers that our observation on increased vulnerability of *VPS4B* knockout cells to VPS4A depletion should be confirmed in non-engineered cancer cell line(s) with native loss (or partial loss) of *VPS4B* expression. To identify such cell lines we used datasets from the Dependency Map (DepMap) portal (<https://depmap.org/portal/>) that were updated after the initial submission of our manuscript (May 2019). This portal systematically catalogs genetic vulnerabilities in human cancer models (currently above 600) identified in genome-scale CRISPR/Cas9 and RNAi screens performed as a part of the following projects: Broad's Project Achilles (Broad Institute, USA), Novartis' Project DRIVE (Novartis Institutes for Biomedical Research, Switzerland) and Sanger's Project Score (Wellcome Sanger Institute, UK).

According to this portal, *VPS4A* (as well as *VPS4B*) are "strongly selective genes", meaning that certain cell lines demonstrate distinctive vulnerability to the perturbation of their expression across the panel of over 500 cancer cell lines tested. Importantly, this observation was cross-validated in a number of screens independently of the approach for gene perturbation (CRISPR/Cas9 vs. RNAi) applied. Specifically, in CRISPR/Cas9 screens 141 cell lines out of 625 tested were sensitive to the perturbation of *VPS4A* expression, while in RNAi screens these were 38 lines out of 550 tested. Moreover, genetic characterization of the investigated cell lines revealed that those most vulnerable to VPS4A depletion had a decreased copy number of the *VPS4B* gene (new Fig 2C). Yet another study, published in April 2019, listed *VPS4A* as one of priority cancer drug targets, based on the data from CRISPR/Cas9 screens combined with the genetic characterization within the Sanger's Project Score (Behan et al, Prioritization of cancer therapeutic targets using CRISPR-Cas9 screens. *Nature* 568:511). Altogether, these very recent data lend strong independent support to our conclusion that targeting the VPS4A activity is a promising target for precision oncology.

To identify and validate a cancer cell model with low or null *VPS4B* expression, as requested by the Reviewers, we compared the top dependency scores of cell lines selected in various screens. We assumed that cells with complete or partial loss of *VPS4B* rely on VPS4A activity but probably are not or less sensitive to VPS4B perturbation. Thus, we aimed to fish out cell line(s) with an altered *VPS4B* gene copy number and differential dependency scores for *VPS4A* and *VPS4B*. For

further experiments we chose HOP62 and SNU410 cell lines (lung and pancreatic cancer, respectively). These cell lines had been tested in at least two independent screens, consistently reaching low *VPS4A* dependency score (CERES or DEMETER2 lower than -0.5, where score of -1 corresponds to the median of all common essential genes) and relatively high *VPS4B* dependency score (CERES or DEMETER2 higher than -0.5, the score of 0 is equivalent to a gene that is not essential) (new Fig EV2C). Importantly, in SNU410 cells *VPS4A* was scored among the top 10 preferentially essential genes (genes with the lowest dependency scores among all genes tested in a given cell line) in both CRISPR/Cas9 and RNAi screens according to DepMap. In HOP62 cells, *VPS4A* was identified among top essential genes only in CRISPR/Cas9 screens.

Having selected the two cell lines, we first confirmed the decreased number of *VPS4B* alleles in SNU410 and HOP62 cells (new Fig 2D). We also verified low *VPS4B* protein abundance in lysates of these cells in comparison to other cancer cell lines (new Fig 2E). Finally, we did confirm that RNAi depletion of *VPS4A* in SNU410 and HOP62 cells (new Fig EV2D) suppressed by over 40% their viability (new Fig 2F, G). Moreover, in case of HOP62 cells their clonal growth was also reduced (SNU410 cells did not exhibit clonal growth), as shown in new Fig 2F (right panels).

Cumulatively, we demonstrated that HOP62 and SNU410 are cell models with partial loss of *VPS4B* that exhibit increased dependency on *VPS4A*. These models might be used for further studies on the synthetic lethality between *VPS4* paralogs in cancer.

2. If the authors would like to extrapolate the findings of combined knockdown, these experiments should also be performed in different normal cell lines, preferable in vivo to show the reduced of absence of effects. This is particular important when no specific inhibitors can be developed.

As requested, we analyzed the proliferation rate of two different non-tumor cell lines: CCD-841CoN and CCD-1070Sk (colon epithelium and skin fibroblasts, respectively) upon single and double knockdown of *VPS4* paralogs (Figure 1 provided for the Reviewer below). Both cell lines are diploid and express *VPS4A* and *VPS4B* (new Fig 2E and Figure 1 provided for the Reviewer, A and B, left panels). In both cell lines we obtained very efficient siRNA-mediated silencing of *VPS4B* expression and quite efficient silencing of *VPS4A* (Figure 1 provided for the Reviewer, A and B, left panels). As a result, we confirmed that none of these cell lines were sensitive to single depletion of *VPS4A* or *VPS4B* (similarly to cancer cell lines with unperturbed *VPS4A* and *VPS4B* expression, e.g. HCT116, DLD1) (Fig 1 provided for the Reviewer, A and B, right panels). In turn, simultaneous depletion of both *VPS4* paralogs severely decreased cell viability of normal cells as a consequence of irreplaceable functions of these proteins in maintaining cellular homeostasis. Based on our data, we believe that cells with unperturbed expression of *VPS4A* and *VPS4B* have some surplus of *VPS4* activity that could be diminished (by siRNA or an inhibitor) without negative impact on cell growth. If this is true, then an inhibitor targeting both *VPS4* paralogs when used in chemotherapy would first affect the growth of those cells that have low expression of one of *VPS4* paralogs (in some patients these would be cancer cells with *VPS4B* loss).

By all means, we are aware that our conclusions are drawn on the basis of short-term *in vitro* studies on a limited number of cell lines and as such may not detect potential side effects that would be generated in a complex organism. However, in our opinion, to precisely evaluate the potential benefits and risks of using a *VPS4* inhibitor in therapy, *in vivo* mouse studies should be performed once such an inhibitor is developed.

Figure 1 provided for the Reviewer. Silencing efficiency of VPS4A and VPS4B, and the viability of CCD-841CoN and CCD-1070Sk cells upon depletion of VPS4A, VPS4B and VPS4A+B.

A, B) Left panels, immunoblot analysis of the VPS4A and VPS4B protein abundance in lysates of CCD-841CoN (A) and CCD-1070Sk (B) cells collected 72 h after transfection with control (siCTRL) or VPS4A- or VPS4B-targeting siRNA (siVPS4A or siVPS4B, various oligonucleotide sequences and their combinations were tested). Vinculin – loading control. Right panels, analysis of cell viability of CCD-841CoN (A) and CCD-1070Sk cells (B) assessed 144 h after transfection with siRNA as indicated.

*Data were normalized (averaged value of siCTRL#1 and siCTRL#2 was set as 100) and are means \pm SEM (n=4). Statistical significance was determined by Kruskal-Wallis followed by Dunn's multiple comparison post test. The following groups were compared: siCTRL#1,#2 group vs. siVPS4A#2,#4,#5 group; siCTRL#1,#2 group vs. siVPS4B#1,#2 group; siCTRL#1,#2 group vs. siVPS4A+B group). NS – non-significant ($p \geq 0.05$), *** $p < 0.001$.*

3. It is surprising that the data in the DRIVE database indicate that SK-CO-1 cells are the most sensitive cell line to VPS4A as well as VPS4B depletion. In addition, one can observe a correlation between the viability scores in the cell line panel for VPS4A and B.

Following the request of Referee #2, we removed the data concerning SK-CO-1 cells from the manuscript, instead including the new data on positively verified HOP62 and SNU410 cells (see point 1 above).

Nevertheless, to address the issue raised by the Reviewer, we re-analyzed the screening data available for SK-CO-1 cell line. According to DepMap, VPS4A and VPS4B are among the top

10 preferentially essential genes for this cell line among 8000 genes analyzed in shRNA screens within the Drive project (with DEMETER2 scores -2.08 and -1.76 for *VPS4A* and *VPS4B*, respectively; Figure 2A provided for the Reviewer). A similar strong dependency on both VPS4 paralogs was also observed for some other cell lines in this RNAi-based and other genome-wide CRISPR/Cas9-based projects (Figure 2B provided for the Reviewer).

We can only speculate that the dependency of SK-CO-1 and other cell lines on both VPS4 paralogs may result from unique proteomes of these cells arising from their individual mutagenic history. Possibly, survival and proliferation of these cell lines may require high rates of ESCRT-mediated processes, e.g. endocytosis (for nutrient or growth factor uptake) that in parallel with concomitant loss of compensatory pathway(s), make them highly sensitive to any VPS4 paralog perturbation. However, only dedicated experimental studies can clarify these issues in the future.

Figure 2 provided for the Reviewer. Various dependencies on VPS4A and VPS4B across cancer cell lines.

A, B) Distribution of dependency scores obtained for a panel of cell lines examined in RNAi-based (A) and CRISPR/Cas9-based (B) screens. Images were downloaded from the DepMap portal (<https://depmap.org/portal/>). Dependency scores of cell lines used in our study were marked in black.

4. One other point is the potential correlation between proliferation and VPS4B expression. Is the expression of VPS4B affected by proliferation rate of cell cycle progression?

To our knowledge, there have been no reports on *VPS4B* expression during cell cycle progression. Thus, to answer the Reviewer's question, we analyzed VPS4B and VPS4A protein abundance in HCT116 cells either serum-starved or treated with cell cycle inhibitors. We observed that both G1- and G2/M-arrested cells maintained unchanged VPS4A and B protein levels (Figure 3 provided for the Reviewer), thus we concluded that the expression of VPS4 paralogs is invariable during cell cycle progression.

In parallel, we addressed a related question of how simultaneous knockdown of VPS4 paralogs affected cell cycle progression. To this end, we depleted HCT116 *VPS4B*^{-/-} cells of VPS4A with siRNA for 72 h and analyzed the distribution of their cell cycle phases. We demonstrated that depletion of VPS4A caused G2/M arrest of HCT116 *VPS4B*^{-/-} cells (new Fig EV4C), most probably due to the interrupted mitotic exit and cytokinesis, as it was previously shown by others (Vietri et al, 2015, Nature 522:231; Mierzwa et al, 2017 Nat Cell Biol 19:787).

Figure 3 provided for the Reviewer. Abundance of VPS4A and VPS4B proteins in cell cycle-arrested cells.

A) Left panel, immunoblot analysis of VPS4B and VPS4A protein abundance in lysates of HCT116 cells treated for 24 h with 5-fluorouracil (5-FU, 10 μ g/ml), nocodazole (100 μ g/ml), docetaxol (20 nM), vehicle (0.1% DMSO) or serum-starved. Vinculin – loading control. NT – non-treated. Right panel, densitometry analysis of the abundance of the indicated proteins based on immunoblot images as shown on the left. Normalized VPS4B and VPS4A protein abundance in non-treated (NT) and vehicle-treated (Vehicle) samples was set as 1. Data are mean \pm SEM (n=4). Statistical significance was assessed using one-sample t-test. ns – not significant, $p \geq 0.05$.

B) Example of flow cytometry analysis of the cell cycle phase distribution of HCT116 cells treated with cell cycle inhibitors as indicated in A.

Referee #2

In this manuscript the authors exploit coincidental loss of VPS4B in colorectal cancer as a biomarker to predict and test VPS4A depletion as a synthetic lethal strategy for this subset of cancers. Loss of the two VPS4 paralogues caused synthetic lethality in cell line and xenograft models. The manuscript characterizes the observed synergy with respect to gene expression changes, cell death mechanisms, and the potential to elicit an inflammatory response. Overall, the promise of targeting presumed 'passenger' mutations like VPS4B has been underexploited and this study represents an interesting and important addition to this field. The manuscript is quite thorough, the data quality look good, and the work tests important mechanisms of cell killing and immune stimulation. I have some suggestions to improve the manuscript below.

Referee #2 Remarks for Author

1. On Page 5, the authors propose to use SK-CO-1 cells as a model for a cell line with native loss of VPS4B but then find that the model is not appropriate for this purpose? Why do they include this section at all? It would be better to remove it, or screen other cell lines with natively low or lost VPS4B.

Following the Reviewer's recommendation, in the revised version of the manuscript we removed data for SK-CO-1 cell line. Instead we present the results obtained for other cancer cell lines with the confirmed decreased *VPS4B* gene copy number and low VPS4B protein abundance (new Fig 2C-G). We discuss this issue in detail in response to Referee #1, point 1.

2. The rationale for looking at gene expression changes by RNA-seq is not clear because VPS4-depleted cells would be expected to have defects in secretion or the endomembrane system. The authors seem to observe a gene expression program consistent with dying cells, which is good, but does not add much to the mechanism of lethality. Did the authors look directly at secretory defects in the double-depleted VPS4A/4B cells?

Indeed, we did not explain our rationale for RNA-seq experiments well enough. In new Fig EV4 we now present data showing that double depletion of VPS4A+B causes simultaneous perturbation of two well-established ESCRT-dependent processes: endocytosis and cell cycle progression. Thus, we are convinced that the lethal phenotype of double-depleted VPS4A+B cells arises from the perturbation of several ESCRT-dependent processes causing irreversible loss of cellular homeostasis. However, by performing RNA-seq analysis we aimed to identify further unknown cellular consequences of depleting either single or both VPS4 paralogs. The results of these experiments gave us hints to study e.g. inflammatory or apoptotic signaling, elaborated in the final part of the manuscript.

3. The finding that both caspase-dependent and independent pathways lead to cell killing in the VPS4A/B depletion model is good but I thought could be framed in the text as eliminating potential resistance mechanisms (i.e. upregulation of anti-apoptotic factors seen in some cancers would NOT make them inherently resistant to this approach).

We fully share the interpretation of the Reviewer but obviously we were not clear enough in the initial version of the manuscript. Now, in the revised Discussion we strengthened our conclusions on this issue, as suggested by the Reviewer.

4. It was not clear in what genetic contexts the authors think this approach would be effective. The authors claim that 70% of CRCs have loss of VPS4B, but that only 2% have biallelic loss. Their model systems have a complete VPS4B loss, so would the synthetic lethality only be effective in the 2% with biallelic loss, or would there be some efficacy of VPS4A knockdown in the 70%? I felt that the manuscript should be explicit about this, and perhaps test partial VPS4B loss for the efficacy of VPS4A-knockdown mediated killing. The future clinical impact and use of VPS4B as a biomarker depends on a clear statement here.

We thank the Reviewer for this important comment. We addressed this issue experimentally in cells with partial loss of *VPS4B*, as also requested by Referee #1 point 1, and these new results allowed us to strengthen the conclusions of our study. Since we confirmed the increased vulnerability of HOP62 and SNU410 cell lines with partial loss of *VPSB* to depletion of VPS4A (new Fig 2 F, G), we believe that a therapeutic approach based on VPS4 inhibitors could also potentially target tumors with incomplete loss of *VPS4B*. However, we are aware that based on our *in vitro* studies with cancer cells harboring incomplete loss of *VPS4B*, it is hard to predict whether and to what extent any inhibition of tumor growth upon VPS4A perturbation would be of therapeutic value. Nevertheless, even if targeting VPS4A were advantageous only for therapy of tumors with a complete loss of *VPS4B* (estimated at 2% of CRC), still a large number of patients could potentially benefit from it (each year there are 1.8 million new cases of CRC as the third most commonly diagnosed cancer according to WHO).

Minor points:

5. On page 10, the authors state that VPS4A and 4B are functionally redundant. I recommend saying 'partly' or 'partially' redundant since their own analysis (e.g. RNA-seq) shows they are not full redundant.

The Reviewer is right. We introduced the suggested phrasing in the text.

6. The grey inset images in Figure 2 are not very useful or easy to see.

The aim of presenting bright-field microscopy images was to strengthen the results of viability assays indicating cell death upon simultaneous VPS4A+B depletion. However, since they were not found useful, we removed them from the new Fig 2.

Figure 2. What is the rationale for using a one-sample T-test?

In our viability assays, we initially normalized growth readout for all transfection conditions to the value of non-targeting control siCTRL#1-transfected cells that was set as 100% in each biological repetition. Having no variation for the normalized control group, we used one-sample t-test to determine statistical significance. However, during the revision, we observed some variability in growth of SNU410 cells transfected with various control siRNA (#1, #2, #3; new Fig

2G) so we decided in each experiment to average values for all non-targeting siRNA and set this average to 100%. To test statistical significance between two groups (siCTRL group vs. siVPS4 group), we used two-tailed unpaired t-test (new Fig 2 F,G). To test statistical significance between more groups (siCTRL group vs. siVPS4 group, siCTRL group vs. siVPS4B, and siCTRL group vs. VPS4A+B group), we used Kruskal-Wallis or ANOVA test (with an appropriate post test) (new Fig 2 A, B). Thus, in the revised Figure 2 we used one sample t-test only to test significance in the colony formation assay for HCT116 cells, because we have no variation in the normalized control group (siCTRL#1, new Fig 2A, middle panel).

Figure 3. I did not understand why the authors showed the right panel on Figure 3b. Are these just the endpoint values? Why is there no indication of statistical significance?

The right panel in Figure 3B presents the endpoint values of tumor sizes and now we state it clearly in the figure legend. Statistical differences between doxycycline treated and untreated groups were presented in the left chart (showing averaged tumor volumes for each group).

2nd Editorial Decision

6 November 2019

Thank you for the submission of your revised manuscript to EMBO Molecular Medicine. We have received the referees' reports, and as you will see the reviewers are now supportive of publication of your study. I am therefore pleased to inform you that we will be able to accept your manuscript pending minor editorial amendments.

Please address referee #1's comments in writing. We would like you to discuss this referee's concerns regarding colorectal cancer versus other cancer types and rephrase as asked. If you do have data at hand, we would be happy for you to include it, however we will not ask you to provide any additional experiments at this stage.

I look forward to reading a new revised version of your manuscript as soon as possible.

***** Reviewer's comments *****

Referee #1 (Comments on Novelty/Model System for Author):

The synthetic lethal interaction between VPS4A and VPS4B has been described before and linked to copy number alterations due to passenger deletion in conjunction with SMAD4 (McDonald et al, Cell 2017). This work also showed that VPS4A dependency is associated with VPS4B copy number alterations (McDonald, fig 5C). It is unclear to me how the authors conclude that the paper by McDonald suggests that "another partner from 18q that remained unidentified". The experiments convincingly show the SL interaction in the cell line panel without the need for an unknown player..

The focus of this manuscript is on CRC for which they determine frequent down regulation of VPS4B expression. However, this is not exclusively for CRC as it also occurs in lung, pancreas and other tissues. The validation of the dependency on VPS4A in VPS4B low cell lines is subsequently performed with a lung and pancreas cell line and not a CRC cell line. As consequence the conclusions drawn from this work with respect to CRC are more correlative than causative with no example of a CRC cell line that has lost or reduced VPS4B expression.

Referee #1 (Remarks for Author):

The authors have responded adequately to the reviewers comments with respect to the use of normal cell lines and the inclusion of cell lines characterized by the loss of VPS4B expression making use of existing databases. However, based on these additional data, it is no longer appropriate to conclude that CRC is particular dependent on this paralog SL interaction. The cell line examples are derived from lung and pancreatic cancer, two tumor types that also seem to most strongly show this SL interaction in Depmap. It is difficult to extrapolate the observations on the additional cell lines as they were selected "a priori" on their dependency on VPS4A and loss of VPS4B expression. One could ask if a set of CRC cell lines with low VPS4B would be chosen without the information on VPS4A dependency, the same result would be obtained.

In order to publish this work, either the authors include CRC cell lines in their analysis or they rephrase the title and other statements about the dependency of CRC on VPS4A to a more general statement about different cancers. They should also correct wording of the reference to the work of McDonald as this interaction was clearly identified and described.

Referee #2 (Comments on Novelty/Model System for Author):

As before I felt the article was technical sound and novel. The study is pre-clinical and so the medical impact is just hard to determine at this point. I do not view this as an impediment to publication but I do not know the EMBO MOL MED mandate.

Referee #2 (Remarks for Author):

I was happy to see a thorough and thoughtful revision of the manuscript and complete responses to reviewer comments. I have no further concerns.

2nd Revision - authors' response

4 December 2019

***** *Reviewer's comments* *****

Referee #1 (Comments on Novelty/Model System for Author):

The synthetic lethal interaction between VPS4A and VPS4B has been described before and linked to copy number alterations due to passenger deletion in conjunction with SMAD4 (McDonald et al, Cell 2017). This work also showed that VPS4A dependency is associated with VPS4B copy number alterations (McDonald, fig 5C). It is unclear to me how the authors conclude that the paper by McDonald suggests that "another partner from 18q that remained unidentified". the experiments convincingly show the SL interaction in the cell line panel without the need for an unknown player.

The focus of this manuscript is on CRC for which they determine frequent down regulation of VPS4B expresion. However, this is not exclusively for CRC as it also occurs in lung, pancreas and other tissues. The validation of the dependency on VPS4A in VPS4B low cell lines is subsequently performed with a lung and pancreas cell line and not a CRC cell line. As consequence the conclusions drawn from this work with respect to CRC are more correlative than causative with no example of a CRC cell line that has lost or reduced VPS4B expression.

Referee #1 (Remarks for Author):

The authors have responded adequately to the reviewers comments with respect to the use of normal cell lines and the inclusion of cell lines characterized by the loss of VPS4B expression making use of existing databases. However, based on these additional data, it is no longer appropriate to conclude that CRC is particular dependent on this paralog SL interaction. The cell line examples are derived from lung and pancreatic cancer, two tumor types that also seem to most strongly show this SL interaction in Depmap. It is difficult to extrapolate the observations on the additional cell lines as they were selected "a priori" on their dependency on VPS4A and loss of VPS4B expression. One could ask if a set of CRC cell lines with low VPS4B would be chosen without the information on VPS4A dependency, the same result would be obtained.

In order to publish this work, either the authors include CRC cell lines in their analysis or they rephrase the title and other statements about the dependency of CRC on VPS4A to a more general statement about different cancers. They should also correct wording of the reference to the work of McDonald as this interaction was clearly identified and described.

The Referee #1 wrote *"This work also showed that VPS4A dependency is associated with VPS4B copy number alterations (McDonald, fig 5C). It is unclear to me how the authors conclude that the paper by McDonald suggests that "another partner from 18q that remained unidentified". the experiments convincingly show the SL interaction in the cell line panel without the need for an unknown player.* We apologize for our unfortunate phrasing when citing the paper by McDonald and colleagues (doi: 10.1016/j.cell.2017.07.005). In their impressive large-scale shRNA screen the

authors found that some cancer cell lines were sensitive to VPS4A depletion and they linked this phenotype to an altered *VPS4B* copy number in these cells. However, the authors did not confirm this finding experimentally. Neither did they analyze the VPS4A/VPS4B protein abundance in sensitive and non-sensitive cells. For these reasons, we considered a synthetic lethal interaction between VPS4A and VPS4B proposed by McDonald and colleagues as a hypothesis that requires further dedicated experimental verification. By writing about a “partner from 18q that remained unidentified” we meant an unconfirmed and uncharacterized interaction of VPS4A with a gene from 18q. However, as rightly noticed by the Referee #1, this statement misrepresented the conclusions drawn by McDonald and colleagues. We corrected it in the final version of the manuscript. This paragraph now reads: “A large-scale screening for cancer vulnerabilities within the Sanger’s Project Score (Behan et al, 2019) and the DRIVE project (McDonald et al, 2017) revealed that some cancer cell lines are very sensitive to perturbed VPS4A expression. The authors of the latter report suggested the existence of a synthetic lethality between VPS4A and VPS4B, however this hypothesis has not been experimentally verified.”

Second, the Referee #1 wrote: “**However, based on these additional data, it is no longer appropriate to conclude that CRC is particular dependent on this paralog SL interaction (..)**“ and “**In order to publish this work, either the authors include CRC cell lines in their analysis or they rephrase the title and other statements about the dependency of CRC on VPS4A to a more general statement about different cancers**”. We respectfully disagree with the Referee. At no place in our paper did we claim that *VPS4B* loss is a unique feature of CRC. Neither did we narrow down the potential application of VPS4A+B synthetic lethality to CRC. Based on the findings from Fig 1B and 1C, we chose CRC as a cancer model with frequent deletions of *VPS4B*. Consistently, by using CRC cancer patient samples we confirmed the downregulation of VPS4B mRNA and protein levels. Subsequently, by using CRC cell lines we characterized the transcriptional and biochemical consequences of simultaneous depletion of VPS4A+B and analyzed its paracrine impact on macrophages. CRC was our only thoroughly characterized model and in our opinion, this entitles us to preserve the proposed title. Generalizing it to different types of cancer would be an overstatement unjustified by our data.

We wish to stress that although most of our work was performed in the CRC model, we were admitting in the text that other cancers may show a similar dependency. For example, in the abstract: “Here, we report that VPS4B gene, encoding an ATPase involved in ESCRT-dependent membrane remodeling, is such a passenger gene frequently deleted in many cancer types, notably in colorectal cancer (CRC).” In the discussion: “Importantly, our demonstration of synthetic lethality between druggable VPS4 paralogs provides a rationale to develop novel therapies targeting VPS4A activity in cancers with 18q deletion, such as CRC.”

However, we followed the Reviewer’s suggestion to emphasize this issue even further, so we have also rephrased other sentences in the discussion. They now read: “Here, by demonstrating the synthetic lethal interaction between two ubiquitously expressed human paralogs *VPS4A* and *VPS4B*, we uncovered a novel therapeutic target to treat patients bearing *VPS4B*-deficient cancers, for example CRC used as a model in our study. (...) Third, we demonstrated that various genetic backgrounds of cancer cell lines did not reverse the synthetic lethality between *VPS4* paralogs (Fig 2). Finally, we showed that the synthetic lethality between *VPS4A* and *VPS4B* is conserved across tumor types (CRC, lung, pancreas; Fig. 2) and species (Fig EV5). (...) In summary, our findings establish a foundation for future work aiming to develop a VPS4 inhibitor as a putative therapeutic for precision therapy of *VPS4B*-deficient cancers such as CRC”.

Corresponding Author Name: Marta Miaczyńska

Manuscript Number: EMM-2019-10812